# Physical limits of flow sensing in the left-right organizer

Rita R Ferreira[1,2,3,4], Andrej Vilfan[5]*, Frank Jülicher[6], Willy Supatto[7]*[†], Julien Vermot[1,2,3,4]*[†]

[1]Institut de Génétique et de Biologie Moléculaire et Cellulaire, Illkirch, France; [2]Centre National de la Recherche Scientifique, Illkirch, France; [3]Institut National de la Santé et de la Recherche Médicale, Illkirch, France; [4]Université de Strasbourg, Illkirch, France; [5]J. Stefan Institute, Ljubljana, Slovenia; [6]Max-Planck-Institute for the Physics of Complex Systems, Dresden, Germany; [7]Laboratory for Optics and Biosciences, Ecole Polytechnique, Centre National de la Recherche Scientifique (UMR7645), Institut National de la Santé et de la Recherche Médicale (U1182) and Paris Saclay University, Palaiseau, France

**Abstract** Fluid flows generated by motile cilia are guiding the establishment of the left-right asymmetry of the body in the vertebrate left-right organizer. Competing hypotheses have been proposed: the direction of flow is sensed either through mechanosensation, or via the detection of chemical signals transported in the flow. We investigated the physical limits of flow detection to clarify which mechanisms could be reliably used for symmetry breaking. We integrated parameters describing cilia distribution and orientation obtained in vivo in zebrafish into a multiscale physical study of flow generation and detection. Our results show that the number of immotile cilia is too small to ensure robust left and right determination by mechanosensing, given the large spatial variability of the flow. However, motile cilia could sense their own motion by a yet unknown mechanism. Finally, transport of chemical signals by the flow can provide a simple and reliable mechanism of asymmetry establishment.

*For correspondence: andrej. vilfan@ijs.si (AV); willy.supatto@ polytechnique.edu (WS); julien@ igbmc.fr (JV)

[†]These authors contributed equally to this work

## Introduction

Chirality describes the asymmetry between an object and its mirror image. How chiral asymmetries arise in physics and biology is a fundamental question that has fascinated researchers for many years (*Morrow et al., 2017*; *Wagnière, 2007*). A number of independent mechanisms of chirality establishment at multiple scales have been identified in living systems, ranging from subcellular with the establishment of chiral cortical flows (*Naganathan et al., 2014*) to the cellular scale during the process of cell fate specification (*Gómez-López et al., 2014*) and to the tissue scale during the process of left-right (LR) axis specification (*Blum et al., 2014b*; *Coutelis et al., 2014*; *Dasgupta and Amack, 2016*; *Hamada and Tam, 2014*; *Levin, 2005*). Similar concepts also exist in the plant kingdom, suggesting widespread mechanisms of chirality transfer (*Wang et al., 2013*). Biological symmetry breaking is often associated with the internal handedness or chirality of molecular motors (*Ferreira and Vermot, 2017*; *Inaki et al., 2016*; *Naganathan et al., 2016*). One of the most striking examples of biological symmetry breaking is the mechanism of LR axis determination in the developing embryo, which is crucial for the asymmetric internal organ positioning. In most of the vertebrate species, LR axis determination is set by a symmetry biasing event which is under the control of a directional flow generated by the chiral beating pattern of motile cilia (*Hirokawa et al., 2009*; *Nakamura and Hamada, 2012*). To date, even though the molecular mechanisms of LR axis determination are highly

conserved from fish, amphibians to mammals (*Blum et al., 2014a*), many steps of this process are not yet understood (*Ferreira and Vermot, 2017*; *Shinohara and Hamada, 2017*; *Wolpert, 2014*).

Prevailing models of LR specification in most vertebrates (*Blum et al., 2009*; *Gros et al., 2009*) involve groups of cells (the LR organizer, LRO) within the presomitic mesoderm (the segmental plate) coordinating asymmetry establishment through the control of a slow-moving flow (the nodal flow) (*Nonaka et al., 1998*) and an intercellular amplification of the asymmetric signals within and around the LRO cells (*Nakamura et al., 2006*). This slow-moving flow is produced by rotation of multiple, motile cilia located at the cell surface of the LRO. This flow leads to a collective cell response that occurs specifically on the left embryonic side of the LRO and is associated with an asymmetric intra-cellular calcium release (*McGrath et al., 2003*; *Yuan et al., 2015*). The LRO is a transient structure and its function is required for a limited period of time. In zebrafish, the LRO is visible between 10 hr after fertilization (3-somite stage) and 16 hr after fertilization (14-somite stage) (*Essner et al., 2005*; *Yuan et al., 2015*).

Cilia generate a directed flow by beating in a spatially asymmetric fashion (*Satir and Christensen, 2007*). Their cycle consists of a working stroke, during which a cilium is stretched away from the sur-face in order to move the maximum amount of fluid, and a recovery stroke, during which it swipes along the surface, thus reducing the backflow (*Marshall and Kintner, 2008*). Cilia in the LRO are rel-atively short and their beating pattern resembles rotation along the mantle of a tilted cone (*Hirokawa et al., 2006*; *Nonaka et al., 2005*; *Okada et al., 2005*). This tilt effectively makes one part of the cycle act as a working stroke and the other part as a recovery stroke. Spatial orientation is, therefore, a key functional feature of motile cilia involved in LR symmetry breaking, as it deter-mines the strength and directionality of the induced flow (*Cartwright et al., 2004*). The molecular mechanisms that set this orientation involve elements of the Planar Cell Polarity (PCP) pathway (*Borovina et al., 2010*; *Hashimoto et al., 2010*; *Song et al., 2010*) as well as flow itself (*Guirao et al., 2010*). At the molecular scale, the chirality of biomolecules has been proposed to provide some of the asymmetrical cues for LR symmetry breaking (*Brown and Wolpert, 1990*; *Levin and Mercola, 1998*). This paradigm works well with cilia, as the sense of cilia rotation is deter-mined by the chirality of the structure of their protein building blocks (*Hilfinger and Jülicher, 2008*).

To accurately discriminate left from right, cells need to robustly sense a signal over the noise associated with flow and cilia beat. Two hypotheses have been proposed for asymmetric flow detec-tion. According to the chemosensing hypothesis, the directional flow establishes a LR asymmetric chemical gradient that is detected by signaling systems which leads to LR asymmetric gene expres-sion and cell responses in the LRO (*Okada et al., 2005*). The mechanosensing hypothesis, on the other hand, proposes that the LRO cells can detect the mechanical effects of flow. It has been sug-gested that this mechanosensing is mediated by a particular type of sensory cilia that is able to trig-ger a local, asymmetric response of the so-called crown cells, which are located at the periphery of the node (mouse LRO) (*McGrath et al., 2003*; *Tabin and Vogan, 2003*). In the zebrafish LRO (also called Kupffer's vesicle or KV), the model for cilia-mediated mechanotransduction suggests that sensing cilia are immotile and will activate a cellular response due to physical asymmetries generated by the flow (*Sampaio et al., 2014*). The flow detection apparatus needs to be remarkably efficient, as demonstrated by the observation that, despite wild-type mice having hundreds of motile cilia in their LRO, proper asymmetry establishment occurs even in mutant mice with only two motile cilia in the LRO (*Shinohara et al., 2012*). Similarly, although wild-type zebrafish usually has around 50–60 motile cilia in the KV, correct asymmetry establishment occurs in zebrafish mutants with only 30 motile cilia (*Sampaio et al., 2014*). Considering that motile cilia might be chemosensors (*Shah et al., 2009*), LR symmetry breaking could rely on a combination of both mechanosensory and chemosensory mechanisms that could work in parallel. To date, however, several issues with both hypotheses have emerged: the molecular nature of a possible diffusing compound involved is still not known (*Freund et al., 2012*; *Shinohara and Hamada, 2017*) and cilia-mediated mechanotrans-duction in the mouse node has recently been experimentally challenged (*Delling et al., 2016*). In many aspects, these debates remain open because the sensitivity of mechanical and chemical detec-tion mechanisms has not yet been assessed quantitatively from a physical standpoint based on prop-erties of cilia and the flows they generate.

To test the sensitivity of the detection mechanisms, we investigated the physical limits of the sys-tem to discriminate left and right in the presence of flow irregularities and noise. We analyzed the system from the level of individual cilia to the scale of the entire organ using live imaging to

determine the physical features controlling the flow. We used these experimental datasets to calculate the flow in unprecedented detail and assess its robustness. We show that the flow velocities and their local variability impose crucial constraints on potential mechanosensing by cilia and investigate the question whether the small number of non-motile cilia can be sufficient to reliably distinguish between the left and the right side of KV. Furthermore, we use the calculated flow profile to simulate the directed diffusive motion of chemical signals and determine the limit on particle diffusivity for which the mechanism is reliable. Our results show that the physical limits to the reliability of mechanical sensing of flow are stronger than the ones associated with chemical sensing and suggest that chemosensation is the key mechanism for LR axis determination.

## Results

### Theoretical analysis of cilia-generated flow patterns

In the current models, cilia-mediated left-right (LR) symmetry breaking is driven by a chiral flow pattern. In the zebrafish embryo, the directional flow corresponds to an anti-clockwise rotation around the dorsoventral (DV) axis when viewed from the dorsal pole of the Kupffer's vesicle (KV) (*Figure 1A*) (*Essner et al., 2005*; *Kramer-Zucker et al., 2005*). Motile cilia are the driver of the flow whose directionality mainly depends on two features: cilia density and spatial orientation. Several studies have focused on cilia orientation and density to explain the flow directionality in mouse and fish (*Borovina et al., 2010*; *Cartwright et al., 2004*; *Montenegro-Johnson et al., 2016*; *Nonaka et al., 2005*; *Okabe et al., 2008*; *Okada et al., 2005*; *Sampaio et al., 2014*; *Smith et al., 2008*) to understand the principles underlying flow generation. These studies led to contradicting conclusions by proposing different types of cilia orientation, such as posterior tilt (*Borovina et al., 2010*), dorsal tilt (*Supatto et al., 2008*), or a mix of the two (*Okabe et al., 2008*). To solve this issue, we used a theoretical approach and developed a simplified biophysical model that can be tested in vivo.

In order to gain insight into the mechanism of flow generation, we first use a simplified model, which does not consider the small scale inhomogeneities of the flow around individual cilia. This simplification is valid in the limit in which both the cilia length ($L$) and the characteristic distance between cilia are shorter than the distance at which we observe the flow. The latter distance is characterized by the radius of the KV ($R$). The cilia layer can then effectively be represented by a net slip velocity at the surface (*Vilfan, 2012*). With typical parameters $L$ = 6 μm, $R$ = 35 μm and a characteristic distance between cilia of 10 μm, these conditions are roughly satisfied when observing the flows in the center of the KV. Calculations with a detailed hydrodynamic model shown later (section 'Single vesicle analysis reveals a significant variability between embryos') confirm uniform flows in the center, but they also show a significant velocity variability near the KV surface, which is not captured by the simplified model.

The effective surface-slip velocity, which replaces the individual cilia in the simplified model, can be calculated from the cilia parameters as follows. We consider cilia covering a surface with area density ρ and rotating with the angular velocity ω along the mantle of a cone with a semi-cone angle $\psi$, tilted by the angle $\theta$ in a direction $\vec{e}_t$. The direction of rotation is clockwise as seen from the distal end towards the KV wall. They induce a net flow velocity above the ciliated layer in the direction perpendicular to the tilt direction (*Smith et al., 2008*; *Vilfan, 2012*)

$$\vec{v} = \frac{C_N \omega L^3}{6\eta} \rho \sin(\theta) \sin^2(\psi) \, \vec{e}_n \times \vec{e}_t \qquad (1)$$

where $\vec{e}_n$ is a unit vector normal to the vesicle surface, $\times$ is the vector product, $C_N \approx 1.2\pi\eta$ denotes the drag coefficient and η the fluid viscosity (see *Figure 1C*). This contribution results from the fact that a cilium moves more fluid during its working stroke, when it is further away from the surface, than during the recovery stroke, when it is closer. At the same time, even without a tilt (i.e. when the rotation axis is orthogonal to the cell surface), a rotating cilium induces a rotary flow with the amplitude $\sim 3C_N \omega L^4/(16\pi\eta) \sin^2(\psi)\cos(\psi)$. Although a surface (or a cavity) lined with untilted rotating cilia at a uniform density does not produce a long-range flow, non-uniformities in surface density ρ do lead to an effective slip velocity

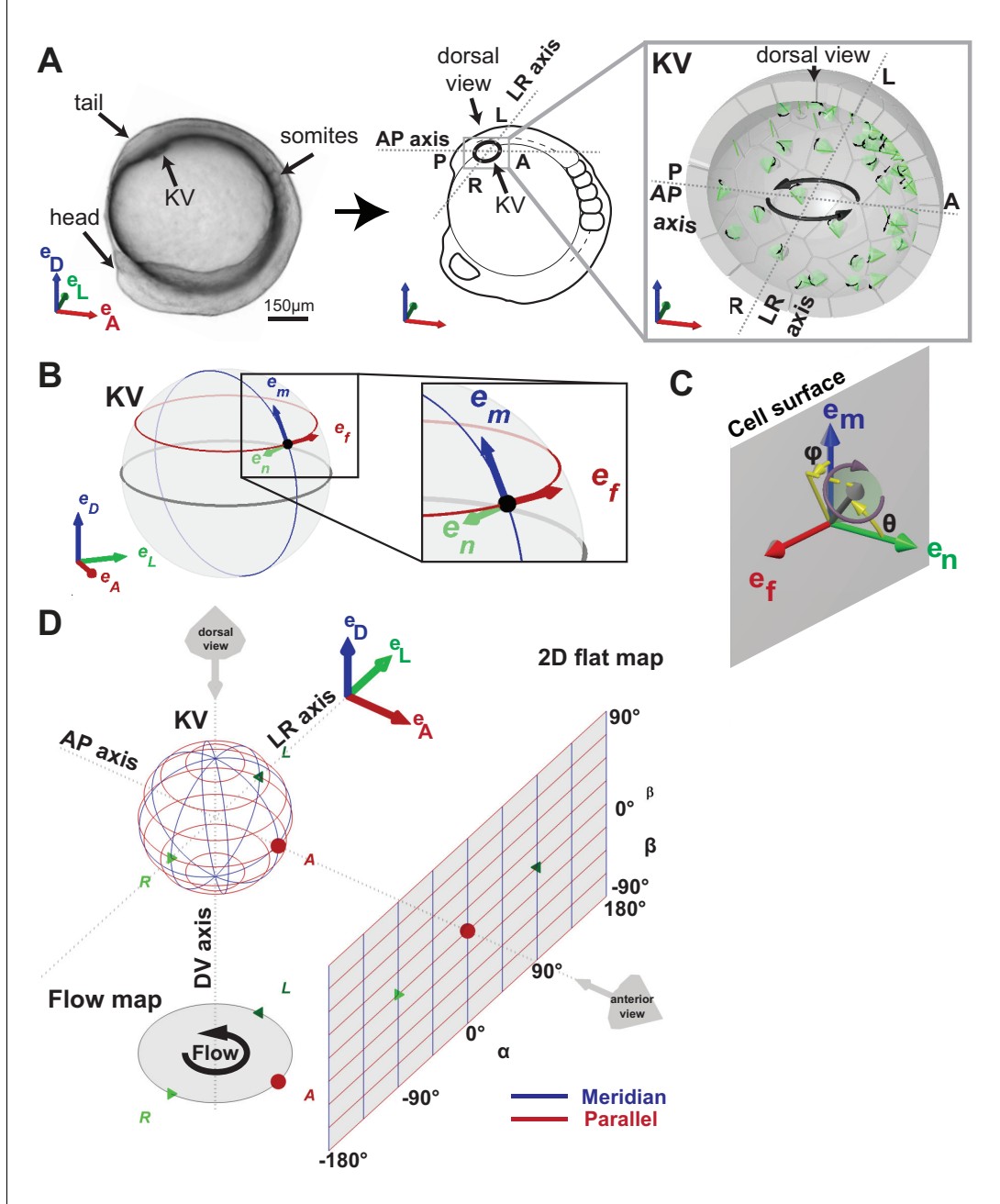

**Figure 1.** Definition of coordinate systems of the Kupffer's vesicle (KV). (A) Side view of a zebrafish embryo at 5-somite stage (left panel) and its schematic drawing (middle panel), highlighting the KV localization (grey box). The zoom-up box (right panel) shows the schematic transverse section of the KV, depicting the cilia (in green), their rotational orientation (black curved arrows) and the directional flow (thick black arrows). (B) $e_m$, $e_n$, $e_f$ are the local basis on the ellipsoid, which are used to define cilia orientation. The vector $e_m$ is aligned along a meridian (blue) from the ventral to the dorsal pole; $e_f$ follows a parallel (red) in the direction of the typical directional flow within the vesicle; $e_n$ is the vector normal to the KV surface and pointing towards the center of the vesicle (green). (C) Cilia 3D orientation is quantified by two angles: θ (tilt angle from the surface normal $e_n$) and φ (angle between the surface projection of the cilia vector and the meridional direction). (D) 2D flat map representation of the KV surface with coordinates α and β. The origin is set in the anterior pole. The embryonic body plan directions are marked as A (anterior), P (posterior), L (left), R (right), D (dorsal) and V (ventral). The body plan reference frame is defined as vectors $e_D$, $e_L$, $e_A$.

The following figure supplement is available for figure 1:

**Figure supplement 1.** Multiscale analysis from individual cilia to 3D modeling of the Kupffer's vesicle (KV).

$$\vec{v} = \frac{C_N \omega L^4}{8\eta} \sin^2(\psi) \cos(\psi) \vec{e}_n \times \overrightarrow{\nabla} \rho \tag{2}$$

The flow observed inside the KV, which is, as a first approximation, characterized by uniform rotation (the fluid moves like a rotating rigid sphere) with the angular velocity $\Omega$ and the surface velocity $\vec{v} = \Omega R \cos(\beta) \vec{e}_f$ (see *Figure 1B* for the definition of the local basis $(\vec{e}_f, \vec{e}_n, \vec{e}_m)$ and *Figure 1D* for the definition of $\beta$) can be achieved in two ways (or a combination thereof):

1. Scenario 1: Dorsoventral gradient of cilia density (*Figure 2A*).
   No cilia tilt, $\theta = 0$, and a density profile $\rho = \rho_0(1 + \sin(\beta))/2$ such that the density reaches its maximum at the dorsal pole.

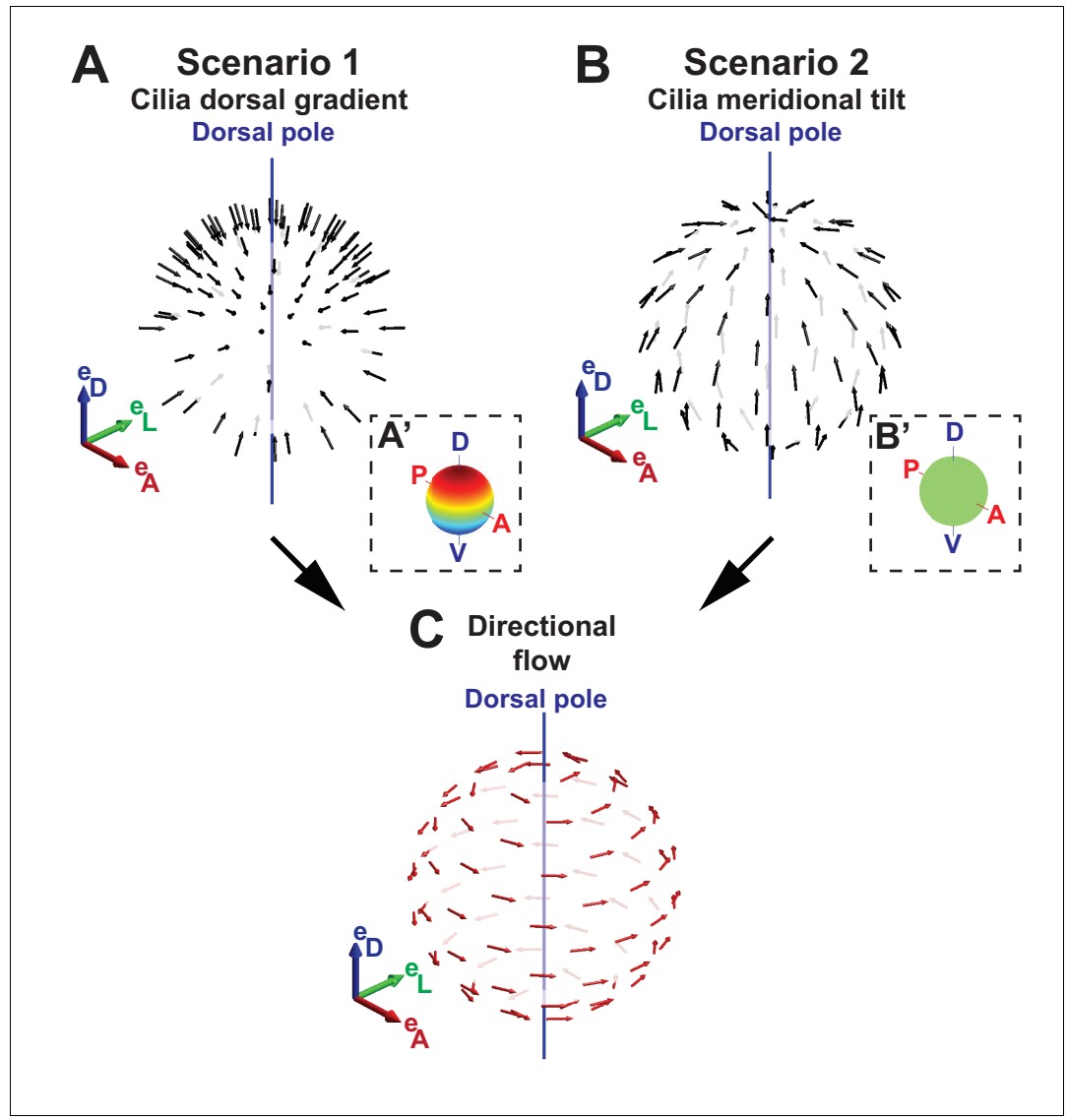

**Figure 2.** Two scenarios for the origin of directional flow. (A) Scenario 1 – dorsal gradient: cilia unit vectors (black) are orthogonal to the surface, but the cilia density increases from the ventral to the dorsal pole. (B) Scenario 2 – meridional tilt: cilia are tilted along the meridians towards the dorsal pole. The insets in (A′–B′) show the density maps on the sphere: a linear dorsal gradient for scenario 1 (A′) and a uniform density for scenario 2 (B′) (color map from blue to red representing low to high cilia density). (C) Both scenarios can theoretically account for the directional flow (red arrows) rotating about the dorsoventral axis observed experimentally. See *Figure 1* for the definition of the body plan reference frame and coordinates systems.

2. Scenario 2: Meridional tilt (*Figure 2B*).
Constant surface density $\rho = \rho_0$ and a meridional tilt $\theta > 0$ obeying $\sin(\theta) = \sin(\theta_0)\cos(\beta)$, i.e., cilia oriented along a meridian, which is the projected line from the ventral to the dorsal pole (blue line in *Figure 1B,D*).

Both scenarios or a combination of both are possible. However, the fluid velocity achieved with scenario 1 is smaller by a factor of $(3L/8R)\cos(\psi)/\sin(\theta_0) \approx 0.1$ as compared to scenario 2. This theoretical analysis shows that the directional flow depends on two main topological features of the vesicle: a profile of cilia density and a cilia orientation pattern following a meridional tilt in the KV.

## 3D-CiliaMap reveals multiple gradients of cilia density

In order to test whether scenario 1 (the cilia density gradient scenario, *Figure 2A*) or 2 (the meridional tilt scenario, *Figure 2B*) leads to the directional flow observed in vivo, we developed a live imaging-based method called 3D-CiliaMap (*Figure 1—figure supplement 1* and *Video 1*). 3D-CiliaMap is designed to quantify experimental features, such as KV size and shape, as well as the spatial distribution, surface density, motility and orientation of cilia (*Figure 1—figure supplement 1*). Developing zebrafish embryos can be accurately staged using the number of somites (blocks of presomitic mesoderm tissue that regularly form along both sides of the neural tube during the segmentation period) (*Kimmel et al., 1995*) (*Figure 1A*). We performed our analysis between 8- and 14- somite stages (SS), when the directional flow in the KV is well established (*Essner et al., 2005*; *Kramer-Zucker et al., 2005*; *Long et al., 2003*; *Lopes et al., 2010*). We analyzed the cilia density in 3D because it has previously been reported to vary along the anteroposterior (AP) axis (*Borovina et al., 2010*; *Kreiling et al., 2007*; *Okabe et al., 2008*; *Supatto et al., 2008*; *Wang et al., 2011*, *2012*). In order to extract average features, the density maps from 20 vesicles with a total of 1197 cilia were averaged and represented either on the average vesicle spheroid (*Figure 3A*) or on a 2D flat map (*Figure 3B*). We found that the average density of cilia did not display any significant differences between left and right sides of the KV (*Figure 3B*). As expected from previous studies (*Wang et al., 2011*, *2012*), a steep density gradient was observed along the AP axis, with increasing density towards the anterior pole of the vesicle (about four times denser than posterior) (*Figure 3A,B*). Interestingly, a gradient of cilia density was also seen along the DV axis, albeit shallower (dorsal being two times denser than ventral) (*Figure 3B*). Together, these results rule out scenario 1, since the dominant AP density gradient would lead to a significant contribution to directional flows around the AP axis, which are not observed (*Supatto et al., 2008*).

## KV motile cilia exhibit a meridional tilt

To test the scenario 2 (the meridional tilt scenario, *Figure 2B*) in vivo, we mapped cilia orientation in embryos between 8- and 14-SS. We defined the orientation of a cilium with a unit vector along the axis of the conical cilia movement and decomposed it in a local orthogonal basis $(\vec{e}_f, \vec{e}_n, \vec{e}_m)$ on the KV surface as defined in *Figure 1B*, where:

1. θ (tilt) is the angle of the cilium with respect to the KV surface normal (0° for a cilium

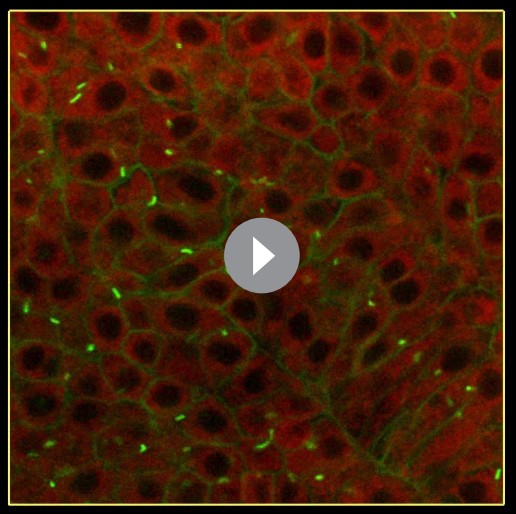

**Video 1.** Visualization of the 3D-CiliaMap processing workflow from 3D live imaging to Kupffer's vesicle (KV) surface segmentation and cilia vectorization. Firstly, one embryo from the *Tg* (*actb2:Mmu.Arl13b*-GFP) (*Borovina et al., 2010*) line soaked for 60 min in Bodipy TR (Molecular Probe) is imaged using 2-photon excitation fluorescence (2PEF) microscopy at 930 nm wavelength. A full z-stack of the KV can be seen. Subsequently, using Imaris (Bitplane Inc.), z-stacks are rendered into 3D volume and the KV cell surface is manually segmented so that only cilia at the cell surface surrounding the KV are visible. At the end, we show the cilia vectorized in the whole volume imaged after 2PEF acquisition.

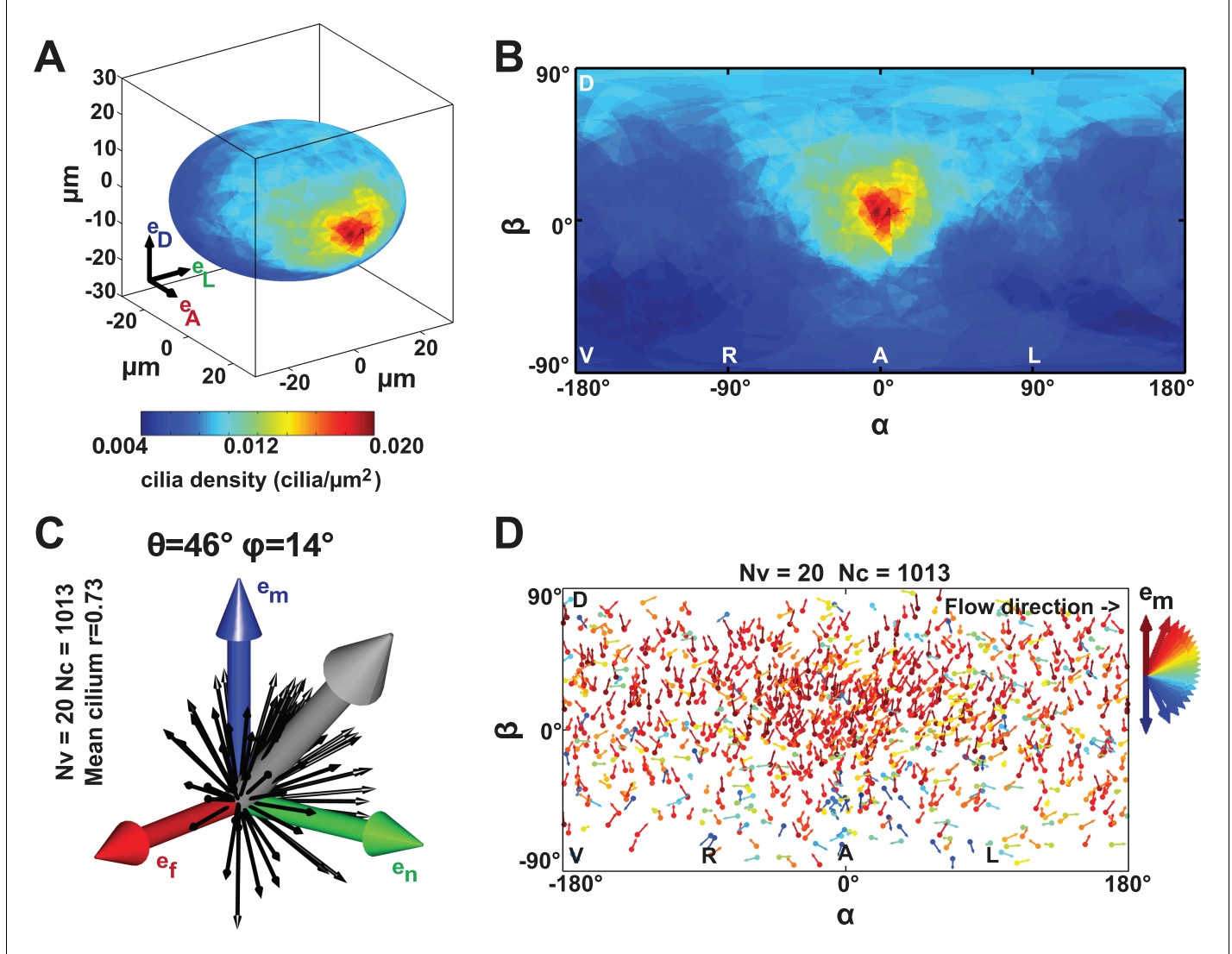

**Figure 3.** Anterior gradient of cilia density and cilia meridional tilt in the Kupffer's vesicle (KV) at 8- to 14-somite stage (SS). (A–B) Averaged cilia density obtained from 20 vesicles represented on a 3D KV map (A) or on a 2D flat map (B) revealing a steep density gradient along the anteroposterior (AP) axis and the resulting enrichment at the anterior pole (in red). (B) Besides the enrichment at the anterior pole ($\alpha = 0°$, $\beta = 0°$), a density gradient along the dorsoventral (DV) axis is also visible ($\beta \geq 0°$ vs $\beta \leq 0°$). (C) Orientations of the 1013 motile cilia analyzed in the local basis ($e_m$, $e_n$, $e_f$) on the ellipsoid: the grey vector (not to scale) shows the vector average of all motile cilia orientations ($\theta = 46°$ and $\varphi = 14°$; $r$=0.73); for the sake of clarity, only cilia orientations from one representative vesicle are shown by black vectors. (D) Cilia orientations ($\varphi$ angles) on a 2D flat map. The majority of cilia point in the meridional direction ($e_m$ in red). $N_v$ = number of vesicles; $N_c$ = number of cilia; $r$ = resultant vector length.

orthogonal to the KV surface and 90° for parallel, *Figure 1C*).

2. $\varphi$ angle is the orientation of the cilium projected on the KV surface (0° for a cilium pointing in a meridional direction towards the dorsal pole, *Figure 1C*).

A meridional tilt would then correspond to $\theta > 0°$ and $\varphi$ close to 0°. Among the 1197 cilia in 20 vesicles, we could determine their orientation and motility status for 86% and 89% of them, respectively. Less than 5% of the cilia were immotile at these stages. Together, we quantified the orientation of 1013 motile cilia (corresponding to 85% of all cilia). We plotted cilia unit vectors in the same local basis (black arrows in *Figure 3C*) and the average cilium as the 3D mean resultant vector (*Berens, 2009*) (gray arrow in *Figure 3C*). The resultant vector length $r$ quantifies the spherical spread (the closer $r$ is to one, the more cilia are concentrated around the mean direction). Despite a

relatively broad distribution of cilia orientations ($r = 0.76$), we found that the θ tilt of the average cilium is near 46° and the average cilium has a 14° φ angle (*Figure 3C*). The resultant vector length and angles ($r$, θ, φ) from individual vesicles are listed in *Table 1*. In total, 65% of all motile cilia exhibited a φ angle between −45° and +45° (meridional quadrant). In addition, no specific spatial distribution of this orientation was detected (red in *Figure 3D*). These data demonstrate that motile cilia are oriented on average with a meridional tilt and support scenario 2 (*Figure 2B*), in which cilia meridional tilt is the dominant mechanism generating the directional flow within the KV (*Figure 2C*). Note that the meridional tilt results in dorsal cilia orientation on the equator and in posterior cilia orientation in the anterior part of the dorsal KV. This might explain why several studies reported different orientation of the cilia in the KV (*Borovina et al., 2010*; *Okabe et al., 2008*; *Supatto et al., 2008*). With

**Table 1.** Statistical properties of all KV analyzed. Table summarizing some of the cilia features collected from the 3D-CiliaMap for individual KV at 3-, 8- and 9–14- somite stage (SS).

| stage | KV number | N cilia | % immotile cilia | Ellipsoid axis a (µm) | b (µm) | Axis ratio a / b | Volume (pl) | Average motile cilium r | θ (°) | φ (°) | Ellipsoid fit RMS residue (µm) | Ω (s⁻¹) |
|---|---|---|---|---|---|---|---|---|---|---|---|---|
| | 1 | 56 | 11% | 27 | 12 | 2.3 | 35 | 0.8 | 37 | 6 | 2.07 | 0.423 |
| | 2 | 23 | 39% | 21 | 12 | 1.8 | 21 | 0.8 | 44 | 2 | 1.56 | 0.207 |
| | 3 | 35 | 3% | 23 | 12 | 1.9 | 27 | 0.8 | 42 | -11 | 2.38 | 0.457 |
| | 4 | 21 | 33% | 18 | 11 | 1.6 | 15 | 0.8 | 22 | 12 | 2.14 | 0.350 |
| | 5 | 19 | 100% | 15 | 9 | 1.7 | 8 | NA | NA | NA | 1.5 | 0.000 |
| 3-SS | 6 | 24 | 54% | 15 | 10 | 1.5 | 9 | 0.7 | 28 | 3 | 1.78 | 0.230 |
| | 7 | 23 | 61% | 14 | 12 | 1.2 | 10 | 0.8 | 18 | -46 | 1.92 | 0.110 |
| | 8 | 40 | 53% | 18 | 11 | 1.6 | 16 | 0.7 | 8 | 22 | 2.05 | 0.034 |
| | mean ± SD | 30 ± 13 | 44% ± 31% | 19 ± 5 | 11 ± 1 | 1.7 ± 0.3 | 17 ± 10 | 0.8 ± 0.1 | 28 ± 13 | -2 ± 22 | 1.9 ± 0.3 | 0.226 ± 0.173 |
| | 1 | 50 | 4% | 29 | 18 | 1.6 | 64 | 0.8 | 37 | -14 | 1.2 | 0.270 |
| | 2 | 49 | 2% | 22 | 8 | 2.8 | 15 | 0.8 | 35 | 6 | 1.1 | 0.595 |
| | 3 | 43 | 0% | 25 | 11 | 2.3 | 30 | 0.7 | 46 | 12 | 2.3 | 0.561 |
| 8-SS | 4 | 50 | 4% | 27 | 12 | 2.3 | 39 | 0.7 | 41 | 16 | 2.5 | 0.329 |
| | 5 | 71 | 0% | 26 | 13 | 2.0 | 36 | 0.8 | 35 | 16 | 1.8 | 0.469 |
| | 6 | 47 | 0% | 30 | 22 | 1.4 | 85 | 0.7 | 21 | 39 | 1.7 | 0.110 |
| | mean ± SD | 52 ± 10 | 2% ± 2% | 26 ± 3 | 14 ± 5 | 2.0 ± 0.5 | 45 ± 25 | 0.7 ± 0.1 | 36 ± 8 | 13 ± 17 | 1.7 ± 0.5 | 0.389 ± 0.186 |
| | 1 | 89 | 16% | 37 | 31 | 1.2 | 174 | 0.7 | 49 | 12 | 2.1 | 0.271 |
| | 2 | 78 | 6% | 27 | 15 | 1.8 | 47 | 0.7 | 46 | 18 | 2.0 | 0.600 |
| | 3 | 80 | 3% | 33 | 20 | 1.7 | 93 | 0.8 | 54 | 13 | 2.2 | 0.412 |
| | 4 | 50 | 0% | 33 | 20 | 1.7 | 91 | 0.7 | 47 | 14 | 2.6 | 0.290 |
| | 5 | 78 | 6% | 31 | 17 | 1.8 | 70 | 0.7 | 36 | 20 | 2.0 | 0.393 |
| | 6 | 94 | 4% | 36 | 21 | 1.7 | 111 | 0.8 | 52 | 10 | 1.7 | 0.424 |
| | 7 | 62 | 11% | 34 | 26 | 1.3 | 124 | 0.8 | 46 | 11 | 1.8 | 0.216 |
| | 8 | 39 | 3% | 24 | 20 | 1.2 | 48 | 0.7 | 54 | 36 | 4.4 | 0.378 |
| | 9 | 35 | 0% | 31 | 13 | 2.4 | 52 | 0.8 | 50 | 10 | 2.6 | 0.245 |
| 9-14-SS | 10 | 49 | 4% | 32 | 17 | 1.9 | 74 | 0.8 | 53 | 2 | 2.3 | 0.275 |
| | 11 | 50 | 2% | 30 | 22 | 1.4 | 82 | 0.9 | 64 | 3 | 2.9 | 0.433 |
| | 12 | 55 | 2% | 30 | 23 | 1.3 | 85 | 0.8 | 61 | 11 | 2.6 | 0.388 |
| | 13 | 49 | 4% | 33 | 10 | 3.3 | 47 | 0.7 | 40 | 32 | 2.1 | 0.224 |
| | 14 | 79 | 5% | 27 | 19 | 1.4 | 56 | 0.7 | 51 | 22 | 1.8 | 0.521 |
| | mean ± SD | 63 ± 19 | 5% ± 4% | 31 ± 4 | 19 ± 5 | 1.7 ± 0.6 | 82 ± 36 | 0.8 ± 0.05 | 50 ± 7 | 15 ± 10 | 2.3 ± 0.7 | 0.362 ± 0.114 |

**Table 2.** List of symbols: Quantities and their values with sources where applicable.

| Symbol | Description | From 3D-CiliaMap | Value: standardized vesicle |
|---|---|---|---|
| $\left(\vec{e}_m, \vec{e}_f, \vec{e}_n\right)$ | Cilium's coordinate system | + | |
| $\left(\vec{e}_A, \vec{e}_L, \vec{e}_D\right)$ | KV coordinate system | + | |
| $\alpha$ | Coordinate | + | |
| $\beta$ | Coordinate | + | |
| $\theta$ | Cilium tilt | + | $0-60°$ |
| $\varphi$ | Cilium orientation on the cell surface | + | $0$ |
| $\psi$ | Cilium, semi-cone angle | | $25°$ |
| $\omega$ | Cilium, angular frequency | | $25 \times 2\pi\ \mathrm{s}^{-1}$ |
| $L$ | Cilium, length | | $6\ \mu\mathrm{m}$ |
| $R$ | KV radius | + | $35\ \mu\mathrm{m}$ |
| $a$ | KV ellipsoid, equatorial radius | + | $R$ |
| $b$ | KV ellipsoid, height | + | $R$ |
| $N_c$ | Number of cilia | + | $70$ |
| $\rho$ | Surface density of cilia | + | |
| $\hat{\rho}$ | Normalized surface density of cilia | + | See *Figure 5* |
| $g(\gamma)$ | Cilia distribution, pair correlation | + | See *Figure 5* |
| $\eta$ | Fluid viscosity | | $0.001\ \mathrm{Pa\,s}$ |
| $r_{Stokes}$ | Diffusive particle Stokes radius | | $0.5-10\ \mathrm{nm}$ |
| $D$ | Particle diffusion constant | | $k_B T/(6\pi\eta\ r_{Stokes})$ |
| $\vec{v}(\vec{x})$ | Fluid velocity inside KV | | calculated |
| $\vec{\Omega}$ | Effective flow angular velocity | | calculated |
| $N_{left},\ N_{right}$ | Number of particles captured on the left/right | | simulated |

such a meridional tilt, all motile cilia can contribute to the directional flow around the DV axis, wherever they are located within the KV.

## Variation in cilia surface density over time affects flow amplitude but not its direction

Current models for symmetry breaking are not taking into account the dynamics of cilia spatial distribution during the process. Nevertheless, the process of LR patterning occurs in a dynamic organ (*Compagnon et al., 2014*; *Wang et al., 2012*; *Yuan et al., 2015*), which might be associated with changes in the spatial distribution and orientation of cilia during the course of LR patterning. Previous reports have shown that the first signs of asymmetric cell response in the KV are observed between 3- and 8-SS (*Francescatto et al., 2010*; *Sarmah et al., 2005*; *Yuan et al., 2015*). To obtain accurate information about these dynamics, we map cilia positioning within the KV in three pools of embryos: early (3-SS), mid (8-SS) and late (9–14-SS) (*Figure 4A*). During this developmental time window, the KV size drastically changes with an average volume increasing from 17 pl at early to 45 pl and 82 pl at mid and late stages. The average number of cilia per vesicle also increases from 30 at early, to 52 and 63 at mid and late stages (*Figure 4—figure supplement 1* and *Table 1*). Not surprisingly, the ratio of motile versus non-motile cilia also changes along with the total number, as previously observed by (*Yuan et al., 2015*). The fraction of motile cilia increases with time (44% of immotile cilia per vesicle on average at 3-SS vs. 2% at 8-SS) (*Figure 4—figure supplement 1*, *Figure 4—figure supplement 2A* and *Table 1*). At mid and late stages, we observed a steep gradient

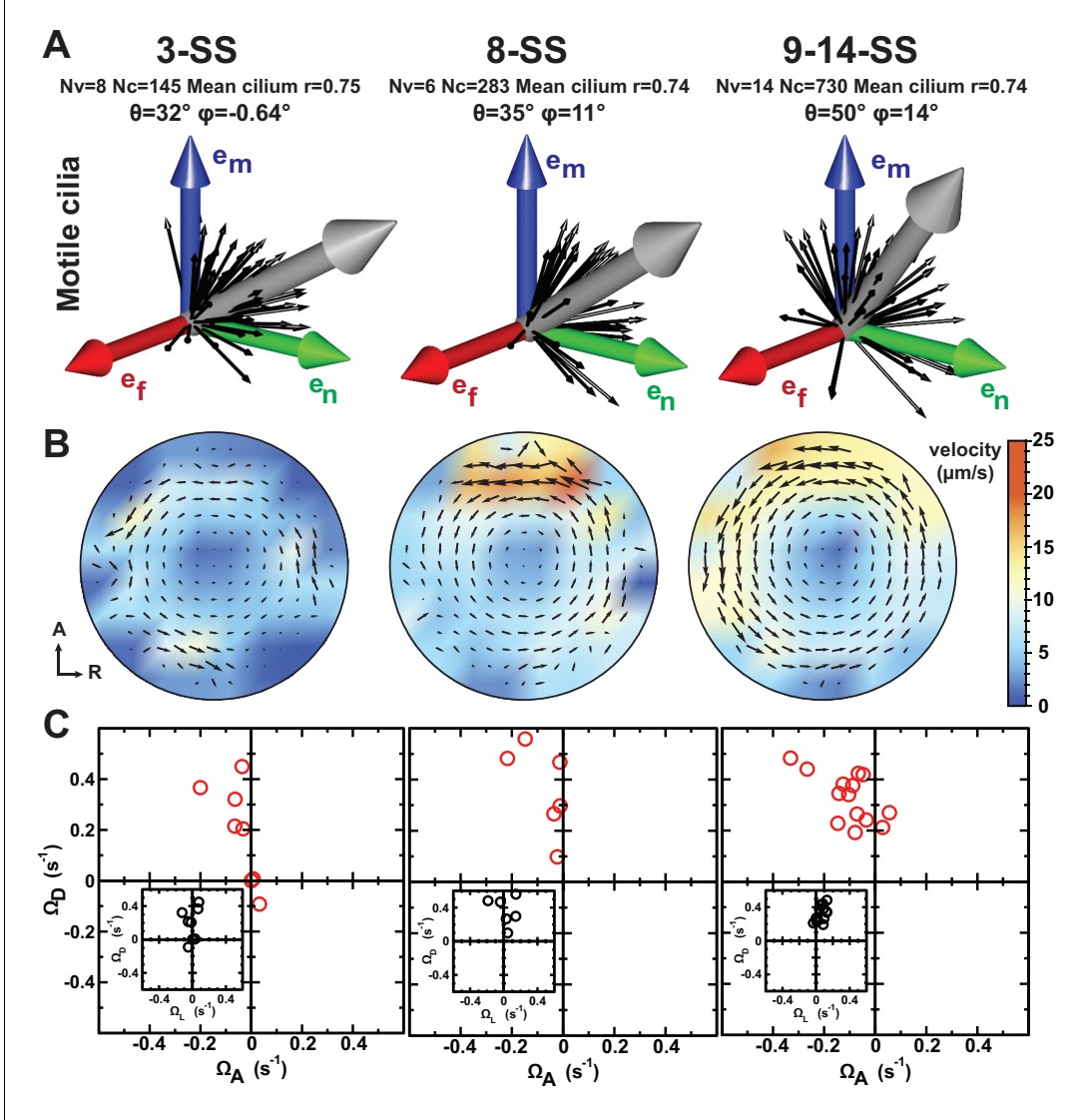

**Figure 4.** Development of flow profiles and cilia orientations over time from 3- to 9–14 somite stage (SS). (A) Cilia orientation in the local basis ($e_m$, $e_n$, $e_f$) over time (see *Figure 3C*). Black vectors show cilia orientations from one representative vesicle. (B) Average flow in the equatorial plane of the Kupffer's vesicle (KV) calculated from cilia maps at each developmental stage. The average flow is rotational about the dorsoventral (DV) axis at all stages, getting stronger anteriorly from 8-SS onwards. A 3D visualization of these flows is shown in *Video 2*. (C) Effective angular velocity ($\vec{\Omega}$) as a measure of rotational flow within a KV over time. Right view of the $\vec{\Omega}$ vector is shown in the main diagrams, posterior view in insets. The following figure supplements are available for *Figure 4*.

The following figure supplements are available for figure 4:

**Figure supplement 1.** Quantification of KV and cilia features comparing the 3-, 8- and 9–14-somite stage (SS).

**Figure supplement 2.** Changes in cilia spatial distribution and orientation over time.

of cilia surface density along the AP axis, which is not yet established at 3-SS (*Figure 4—figure supplement 2B*). Instead, a DV gradient of cilia density is present at 3-SS.

We next investigated the emergence of cilia motility and orientation by comparing early, mid and late stages. We observed that the spread around the average motile cilium is constant over time (resultant vector length *r* = 0.74 or 0.75) (*Figure 4A*). Additionally, it seems that the average

orientation angle θ of the motile cilia increases over time (from 32° to 50°, *Figure 4A*). When focusing on 3-SS embryos, we found that motile cilia at this stage already exhibit a clear meridional tilt, as the angle φ of the average motile cilium is close to 0° (57% of all cilia exhibited a φ angle in the meridional quadrant [−45°, +45°]) and θ is high (32°) (*Figure 4A*). Together, these data suggest that the few motile cilia at 3-SS are already well oriented and generate a flow of low amplitude but in the proper direction.

We suspected that the increasing number of motile cilia and the changes in their spatial distribution would significantly alter the flow profile between 3- and 8-SS. To test this, we used our 3D cilia maps to numerically calculate the flow they generate at the different developmental stages (*Figure 4B* and *Video 2*). We first validated our flow simulation by comparing it with experimentally measured flow profiles along the AP axis (*Figure 5*). As expected, due to the increase in anterior cilia density, the flow amplitude increases over time (*Figure 4B* and *Figure 6A*) with the most pronounced increase in the anterior region. We quantified the directionality of the flow by calculating the effective angular velocity $\vec{\Omega}$, defined as the angular velocity of a uniformly rotating sphere with the same angular momentum as the circulating fluid in the vesicle (*Figure 4C* and *Table 1*). Our results consistently show that in 8-SS and 9–14-SS embryos, the rotational flow persistently points in one direction. Most importantly, in 5 out of 8 vesicles, the flow direction is already set at 3-SS (*Figure 4C* and *Figure 6—figure supplement 1A*), suggesting that directional flow can emerge as early as 3-SS, even though the flow is of low amplitude (*Video 2*). As a consequence, the establishment of an AP gradient of cilia from 3- to 8-SS does not affect the flow direction, which further supports cilia meridional tilt as the dominant mechanism used to generate directional flow within the KV.

## Single vesicle analysis reveals a significant variability between embryos

An additional element to consider in the physical mechanisms of symmetry breaking is its robustness. Given that 90–95% of the zebrafish embryos have a properly positioned left axis (*Gokey et al., 2016*), the mechanism eliciting LR bias has to be highly robust even though the KV size, which affects LR patterning, is variable across embryos (*Gokey et al., 2016*). We sought to directly probe for the robustness of the biophysical features of the cilia through single vesicle analysis. We first analyzed cilia density in individual vesicles and found that cilia density and orientation are very variable from embryo to embryo (*Figure 6B–D*, *Figure 6—figure supplement 1A–C* and *Table 1*). Making use of the cilia maps observed in individual KV and our model of 3D flows, we determined the expected flow profiles. We assessed the general amplitude of the flow and found strong variability in the local flow velocities between individual vesicles at every developmental stage (*Figure 6B–D* and *Figure 6—figure supplement 1A–C*). In particular, the calculated profiles reveal a high level of variability of the difference between left and right flow amplitudes, without a persistent bias (*Figure 6—figure supplement 1A–C*). Similarly, we found the maximum velocity at the anterior and posterior poles of the KV is variable from embryo to embryo (*Figure 6B–D* and *Figure 6—figure supplement 1A–C*). We conclude that the local flow amplitude itself cannot be a good indicator of the embryonic side and is too variable to serve as a robust predictor of the left and right side of the vesicle. By contrast, the strength of the rotational flow characterized by the effective angular velocity $\vec{\Omega}$ appears robust (*Figure 4C*). Thus, while the average vesicle highlights a highly stereotyped organization, single vesicle analysis uncovers a high diversity of densities, orientations and local flow profiles. This identifies the directional flow as the most robust left-right asymmetric feature in the vesicle.

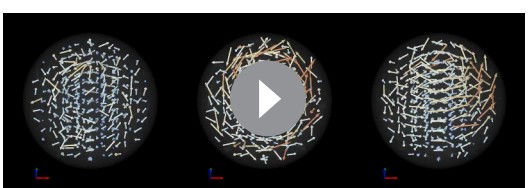

**Video 2.** 3D visualization of the calculated average flows at 3-somite-stage (SS) (left), 8-SS (center) and 9–14-SS (right). See *Figure 4B* for the velocity color scale. The axes show the direction of anterior (red), left (green) and dorsal (blue). At all stages the average flow is directional around the dorsoventral (DV) axis, but the flow velocity increases between 3- and 9–14-SS. The flow profiles in the anteroposterior (AP)-left-right (LR) plane are shown in *Figure 4B*.

## Comparing flow-mediated transport of signaling molecules and mechanical flow sensing as mechanisms for symmetry breaking in the KV

The hypotheses for the mechanisms of LR symmetry breaking are usually divided into two competing classes, independently of the topological differences amongst species: asymmetric distribution of signaling molecules (*Hirokawa et al., 2006*; *Okada et al., 2005*) or asymmetric mechanical influence (*Hamada and Tam, 2014*; *Yoshiba and Hamada, 2014*). Making use of the information gained from 3D-CiliaMap and numerically calculated flow profiles, we developed numerical simulations based on specific arrangements of cilia in order to corroborate or refute different hypotheses for symmetry breaking in the LRO. In particular, we took into account the variability between embryos that we observed experimentally.

### Mechanosensory mechanism 1: directional flow sensing

We first evaluated the '*Directional flow-sensing mechanism*' where the left side is detected by sensing the directionality of the circular flow. Such a mechanism implies that cells need to distinguish between a fluid moving from anterior (A) towards posterior (P) (on the left side) and a fluid moving from P towards A (right side), and the cilium needs to sense the direction of flow. It also has to overcome the following obstacles: (1) The strong temporal variation associated with beating cilia overlays the directional flow to be detected; (2) In addition, the flow fields of adjacent cilia also perturb the static component of the directional flow; (3) The cilia would need a detection threshold sufficiently low to detect the relatively weak flows in the KV. Using our computational modeling, we tested the feasibility of criteria 1–3.

We used the cilia distributions from 20 analyzed vesicles, as well as a larger number of randomly generated vesicles with the same density profiles and interciliary distance distributions (see Materials and methods). In each vesicle, we randomly chose three immotile cilia on each side and calculated the torque acting on them. The torques acting at the base of a cilium are calculated using the numerically determined force distributions along its length. The torque component that could potentially serve as the LR determinant is the meridional one, bending the cilium towards A or P (*Figure 7A*). In order to estimate the fraction of cilia that measure the torque in the correct direction above a certain threshold, we plotted cumulative distributions of the meridional component of the torque vector (*Figure 7B,C*). The instantaneous torques (dotted lines) show a broad distribution and have a direction opposite to that of the directional flow in about 25% of the cases. These fluctuations are caused by the beating of adjacent cilia, which induces on average an oscillating torque with an r. m.s amplitude of $9 \times 10^{-19}$ Nm. Additionally, the torques are of a similar order of magnitude as the thermal fluctuations acting on the cilium, whose r.m.s. amplitude we estimate as $2 \times 10^{-19}$ Nm (Appendix B). A simple estimate shows that both the thermal and oscillatory noise can be suppressed by temporal averaging (low-pass filtering) the signal with a time constant longer than 2 s. The distribution of time-averaged torques is shown by dashed lines in *Figure 7B and C*. A sensitivity threshold of $2 \times 10^{-19}$ Nm would be sufficient to achieve a reliability of 95% (less than 5% of the immotile cilia on the left side would not detect the posterior-directed flow) in vesicles with the generated distributions (*Figure 7B*, dashed lines), but the greater variability of experimentally characterized vesicles does not allow this level of reliability (about 10% of the cilia are subject to flows of the opposite directionality, *Figure 7C*, dashed lines). A sufficient reliability can only be achieved by additionally averaging the torques detected on all three cilia on one side (solid lines). In this case the required detection threshold is $10^{-19}$ Nm. For comparison with other flow sensing cilia, we calculated the uniform shear rate required to exert the same torque. We find that for a 6 μm long cilium the threshold torque corresponds to a shear rate of 0.5 s$^{-1}$ (shear stress 0.5 mPa). In renal cilia, Rydholm and colleagues (*Rydholm et al., 2010*) observed calcium signals with shear stresses of 20 mPa and higher. These results indicate that the mechanosensory detection of flow would require cilia with an ability of direction-sensitive flow detection with a threshold 1–2 orders of magnitude lower than known comparable mechanosensory cilia. Moreover, many KV have fewer than three immotile cilia on each side, the number that would be needed for ensemble-averaging to overcome spatial inhomogeneities of the flow. The number of immotile cilia is higher at 3-SS, but the weaker and less regular flow excludes reliable side detection at that stage (*Figure 7—figure supplement 1*). Together, these results suggest that flow sensing in itself is difficult as the flow is weak and masked

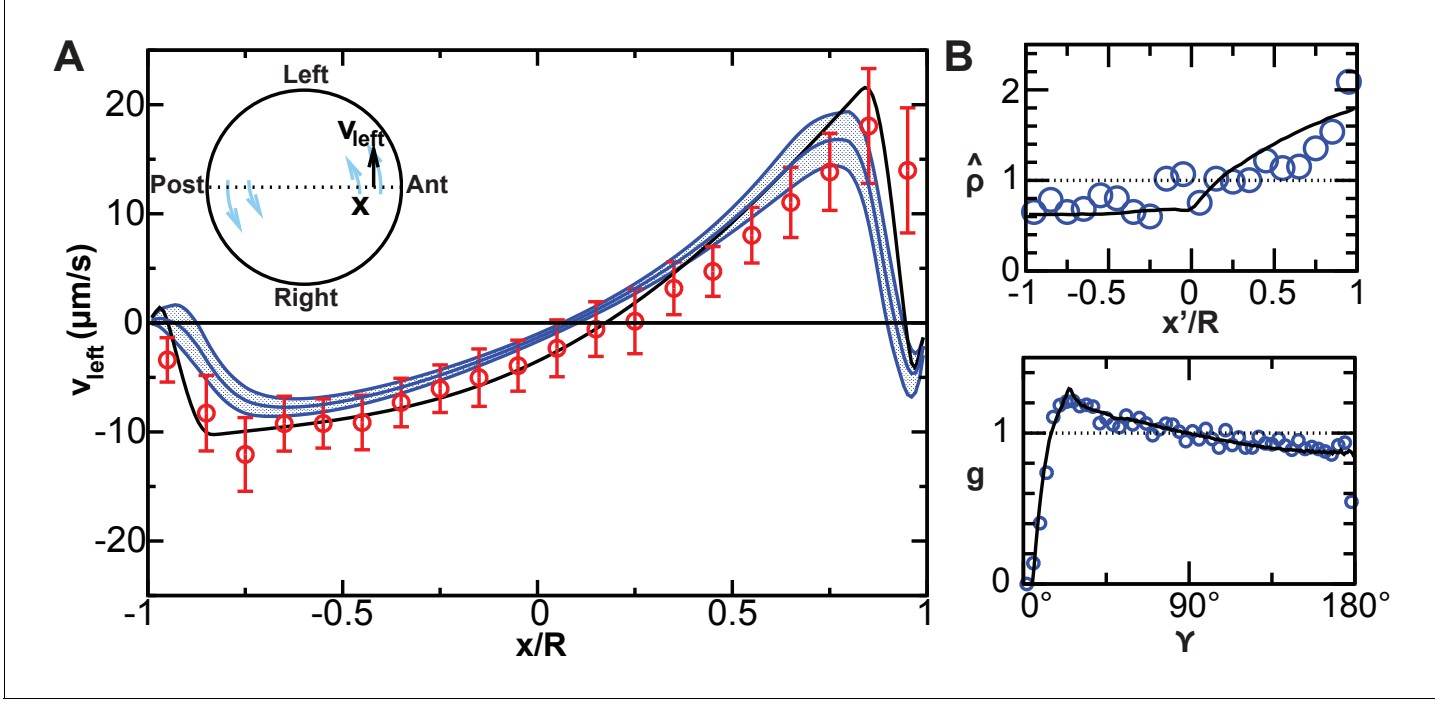

**Figure 5.** Validation of calculated flow profiles. (A) Velocity profile along the anteroposterior (AP) axis (anterior: $x/R = 1$; posterior: $x/R = -1$), positive values indicate leftward flow. Red: experimental values obtained with particle tracking (*Supatto et al., 2008*). Blue: calculated flows using observed cilia distributions from vesicles from 8-somite stage (SS) to 14-SS (mean ± std. error; $N_v = 20$). Black: simulations using randomly generated cilia distributions. (B) Statistical features used to generate cilia distributions (blue circles: experimental distributions, black line: model). Top panel: normalized surface density as a function of the position along the tilted AP axis ($x' = 1$ at the point with maximum density, $(\alpha, \beta) = (0, 15°)$); bottom panel: pair correlation function as a function of the angular distance between two cilia. $N_v$ = number of vesicles.

both by spatial and temporal fluctuations. We thus expect that such a mechanism of sensing would lack the robustness necessary for setting the LR axis accurately.

## Mechanosensory mechanism 2: cells sense the motion of their own cilia

An attractive possibility based on mechanosensing is that cells '*detect the motion of their own cilia relative to already established body axes*'. In this case, the rotating cilium provides a cell the necessary chirality information, while the directional flow in KV would only appear as an epiphenomenon. The tip velocity of a beating cilium (400 μm/s) is significantly higher than the typical flow velocity (10 μm/s), which implies that in a motile cilium the torques caused by its own motion largely surpass those caused by the directional flow. Using the mobility matrix of a model cilium (see Materials and methods) we calculated the torque components acting on the base of an isolated dorsally tilted motile cilium, positioned either on the left or on the right side (*Figure 7D*). The dashed lines show the time averages of the three torque components. The average torque caused by an active cilium's motion is about 20 times higher than the torque caused by the directional flow (*Mechanosensory mechanism 1*). If a cell could discriminate between a torque towards A or P exerted at the base of its motile cilium, the time-averaged meridional component of the torque vector (blue line), which has a magnitude of about $10^{-17}$ Nm, could serve as a side discriminator. Thus, cells sensing the torque direction generated by their own cilia to is a possible mechanism for an asymmetric response in the KV.

## Flow-mediated transport of a signaling molecule

As the third mechanism, we investigated the possibility of '*flow-mediated transport of a secreted signaling molecule*' in the KV. It is known that classical motile cilia also contain receptors to detect the external chemical environment (*Shah et al., 2009*). We propose that cilia on the left and on the

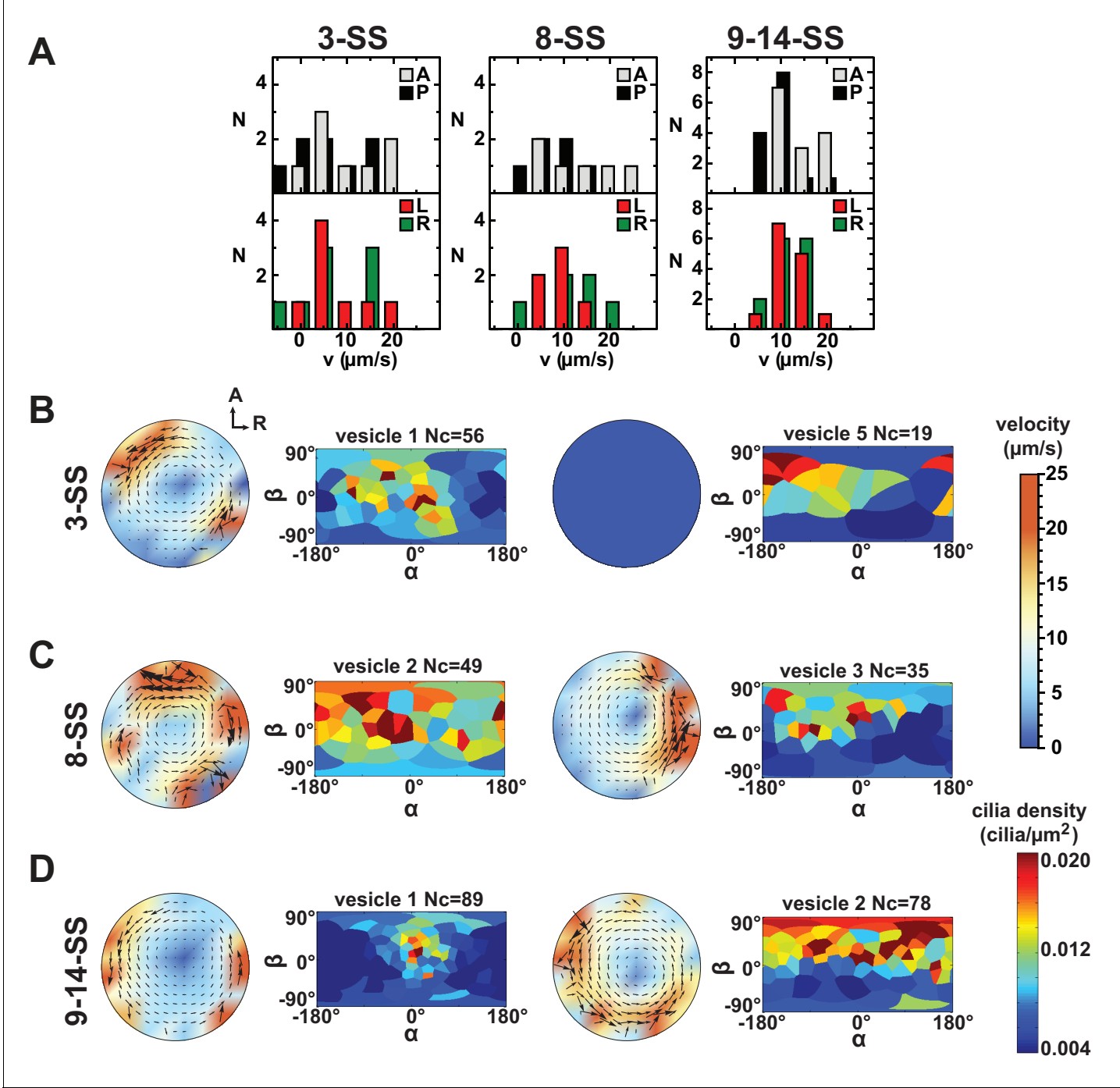

**Figure 6.** Variability in cilia distributions and flow profiles between individual Kupffer's vesicle (KV) at 3-, 8- and 9–14- somite stage (SS): (A) Distributions of flow velocities in individual KV at 3-, 8- and 9–14-SS. The upper panel shows the mean velocities in the regions around the anterior (A) and posterior (P) poles and the lower panel around the left (L) and right (R) poles. (B–D) Flow profiles and 2D cilia density maps for two representative KV at 3-SS (B), 8-SS (C) and 9–14-SS (D) (see *Figure 6—figure supplement 1* for all individual KV).

The following figure supplement is available for figure 6:

**Figure supplement 1.** Flow profiles and 2D cilia density maps for all Kupffer's vesicles (KV) analyzed at 3- somite stage (SS) (A), 8-SS (B) and 9–14-SS (C), showing a great variability between embryos.

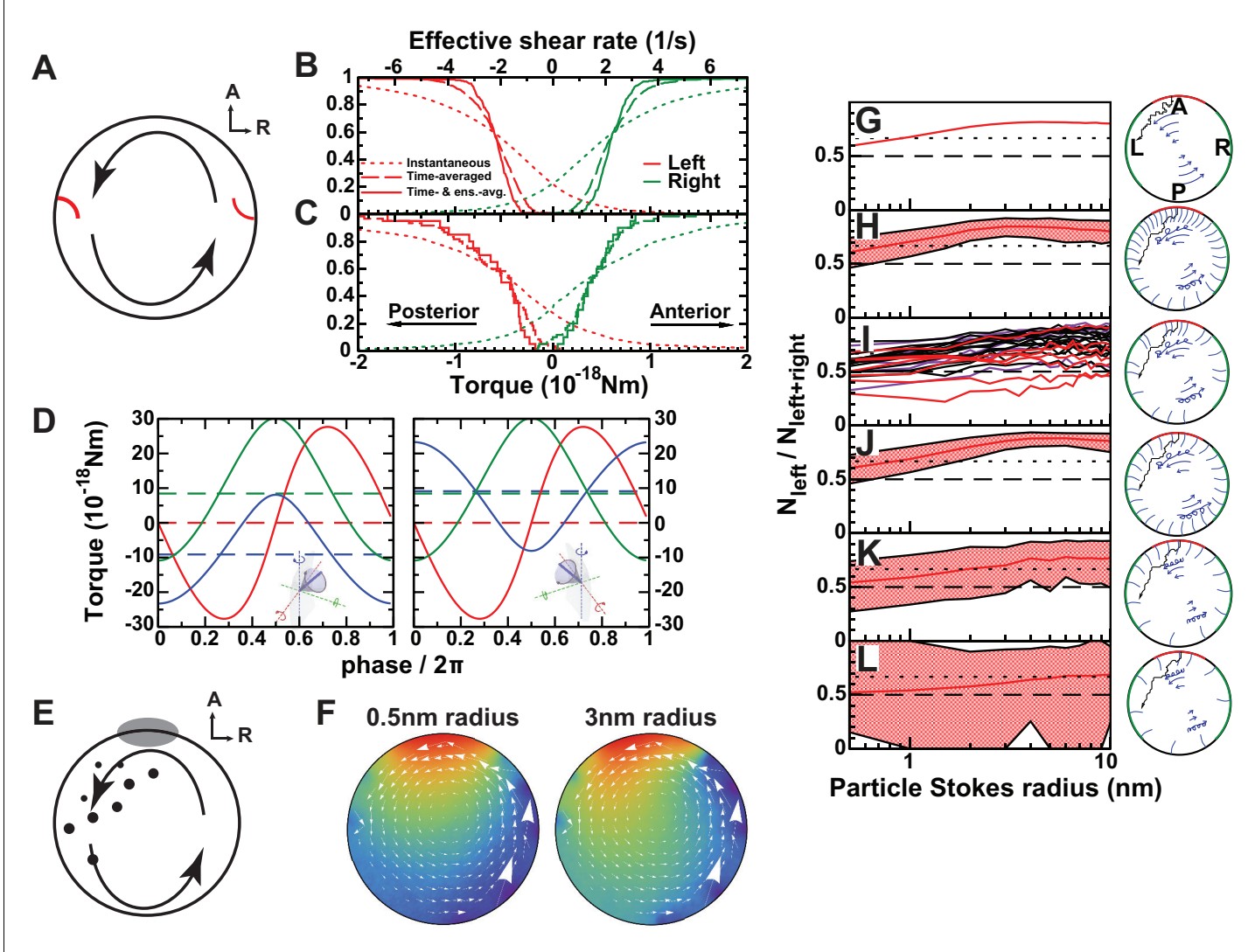

**Figure 7.** Physical limits of possible side detection mechanisms. (A) Mechanosensory mechanism 1: directional flow sensing. Sensory cilia (red) on the left (L) and on the right (R) side are deflected by the rotational flow (arrows). They must be able to distinguish between anterior- and posterior-directed flows. (B–C) Cumulative fraction of cilia with the anterior acting force below (right, green) or above (left, red) the value on the abscissa. The dotted lines show instantaneous values (blurred by oscillatory flows of adjacent cilia), dashed lines show the temporal average, and the continuous line the temporal and ensemble average of 3 immotile cilia on each side. The diagrams show the results on randomly generated (B) and experimentally characterized (C) vesicles. The results show that reliable detection (<5% error) would need a sensitivity threshold of $1 \times 10^{-19}$ Nm. The upper scale shows the effective flow shear rate above a planar surface that induces the equivalent torque on an isolated passive cilium of the same length. (D) Mechanosensory mechanism 2: detection of a cilium's own movement. According to this mechanism, a cell can sense the torque components caused by the motion of its active cilium through the viscous fluid. The lines show the meridional component towards posterior (blue), parallel component towards dorsal (red), and normal component (green). The meridional component shows a temporal average of $10^{-17}$ Nm that could potentially allow discrimination between left (left panel) and right (right panel) side. (E) Chemosensory mechanism, based on flow mediated transport of a signaling molecule. Particles are secreted from a region 30° around the anterior (A) pole and then travel diffusively through the rotating fluid. They get absorbed upon encounter with any cilium outside the anterior region. Eventually, particles absorbed in a 45° region around left-right poles are counted. (F) Average particle concentration (arbitrary units) in the equatorial plane for particles where diffusion dominates fluid circulation (Stokes radius = 0.5 nm, top) and those with drift dominating (3 nm, bottom). In the latter case, an asymmetry in the distribution is clearly visible (*Video 3*). (G–L) Fraction of particles counted on the left among the total count of left and right for different scenarios. The dotted line shows a proposed detection threshold with a left to right ratio of 2:1. The red line shows the average vesicle and the shadowed region the interval between the 5th and the 95th percentile. (G) Continuous model with uniform circulation ($\Omega = 0.5$ s$^{-1}$). (H) Randomly generated cilia distributions with natural parameters. (I) Simulation on individual vesicles at 3-SS (red), 8-SS (indigo) and 9–14-SS (black). (J) Same as H, but homogeneous cilia distribution. (K) Same as H, but reduced number of cilia ($N_c = 35$). (L) Further reduced number of cilia ($N_c = 20$). The following figure supplement is available for *Figure 7*.

*Figure 7 continued on next page*

**eLIFE** Research article

Computational and Systems Biology | Developmental Biology and Stem Cells

*Figure 7 continued*

The following figure supplement is available for figure 7:

**Figure supplement 1.** Cumulative torque distributions on immotile cilia as in *Figure 7C*, but using cilia maps at 3-somite stage.

right side (45° around LR axis, representing the areas where the first asymmetric responses have been observed (*Francescatto et al., 2010*; *Sarmah et al., 2005*), act as detectors that absorb small particles in contact with their surface. Because the first asymmetric signal was observed on the left (*Yuan et al., 2015*), we propose that these particles are secreted in the anterior region (30° around the anterior pole) and that cells in this region do not absorb them (*Figure 7E* and *Video 3*). As a rough estimate, we expect that flow-mediated transport requires a Péclet number $Pe = vR / D > 1$, which states that advection dominates over diffusion. With $\Omega = 0.5\,\mathrm{s}^{-1}$, we get $Pe = \Omega R^2 6\pi\eta r_{\mathrm{Stokes}} / (k_B T) = r_{\mathrm{Stokes}} / 0.4\,\mathrm{nm}$. The condition $Pe > 1$ is thus fulfilled for particles above nanometer size. To assess the feasibility of flow-mediated transport quantitatively, we simulated the diffusion of small particles in the flow fields calculated before (*Figure 7F*). We propose that asymmetry is detected based on the cumulative number of particles detected on each side. Therefore, the time course of particle secretion is not important for our arguments. As a measure of asymmetry, we used the number of particles detected on the left side $N_{\mathrm{left}}$, divided by the total number detected on the left and right combined ($N_{\mathrm{left+right}}$) (*Figure 7G–L*). We estimate that the difference is detectable if the ratio between left and right is at least 2:1, that is, if the fraction is higher than 2/3 (*Figure 7G–L*). Computational results on 20 vesicles are shown in *Figure 7I*. For signaling particles with a Stokes radius of 2 nm, we obtain a robust asymmetric readout in 18 of 20 vesicles. In addition, we performed the computation on a larger number of randomly generated vesicles and see that, for particle sizes of 2–10 nm, more than 95% of the vesicles fulfill the asymmetry requirement. In order to elucidate the requirements on the arrangement of cilia needed for robust asymmetry establishment, we also simulated vesicles with alternative distributions. *Figure 7J* shows a scenario with cilia distributed homogenously around the whole vesicle, which is equally efficient. It is therefore unclear whether the increased cilia density in the anterior region fulfills a purpose with regard to symmetry breaking. It is possible, however, it becomes beneficial for different secretion scenarios. Since it was previously shown that there is a minimum number of motile cilia required to achieve LR patterning in vivo (*Sampaio et al., 2014*), we tested the model with a smaller number of motile cilia. With half the number of cilia ($N_c = 35$, *Figure 7K*) the vesicles still show sufficient average asymmetry, but fail to achieve 95% reliability. Interestingly, the mechanism becomes dysfunctional with 20 motile cilia, which is very close to the minimum observed in vivo by (*Sampaio et al., 2014*) (*Figure 7L*). This indicates that '*flow-mediated transport of a secreted signaling molecule*' is plausible in the KV and that the KV contains enough motile cilia to allow robust symmetry establishment through asymmetric transport of signaling molecules, but not much more than needed.

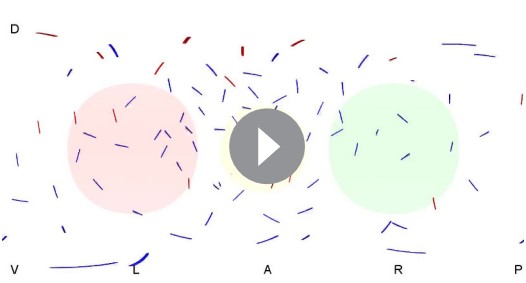

**Video 3.** Simulated transport of signaling molecules in the Kupffer's vesicle (KV). Panoramic view of the KV as seen from the center. The cilia distribution is obtained from vesicle 1 at 9–14-somite stage (*Figure 6—figure supplement 1C*). Cilia shown in blue are motile and those in red immotile or undetermined. Signaling particles ($r_{\mathrm{Stokes}} = 5\,\mathrm{nm}$, not to scale), secreted from anterior (yellow), are subject to Brownian motion biased by the leftward flow until they are absorbed by a cilium (shown yellow after particle capture). 10 s of video represent 1s real time. Link for 360° video: https://youtu.be/1caSzBIe5rA

## Discussion

Here, we have investigated physical mechanisms that could underlie the left-right (LR) symmetry breaking of the vertebrate body plan. We have combined microscopy and large-scale cilia mapping in living animals with theory to analyze the physical limits of the generation and detection of LR asymmetric flows. The experimental analysis of cilia patterns over time allowed us to generate

a comprehensive map of complex in vivo cilia behaviors at key stages of LR symmetry breaking. We used this approach to assess the reliability by which asymmetric cues can be detected by cells in the process of the determination of the LR axis of the vertebrate body plan.

Recent work in fish and mice has highlighted two possible ways to mechanically sense the directional flow: asymmetric flow velocity sensing (*Sampaio et al., 2014*) and/or flow direction sensing (*McGrath et al., 2003*). Our results argue against asymmetric velocity sensing because no robust asymmetries in the flow amplitude were observed in the KV, as previously suggested (*Smith et al., 2014*). The calculated flow pattern obtained when using cilia parameters collected in vivo shows that its directionality is the most robust sign of chirality in the organizer. Another level of complexity, which is shared between mouse and fish, is the presence of beating cilia in the near vicinity of the immotile cilia. It has been suggested that these cilia detect the flow direction (*Yoshiba et al., 2012*). Indeed, the average distance from one cilium to its nearest neighbor is under 8 μm in the KV. At such a distance, the strong local flow perturbations have been shown to be chaotic (*Supatto et al., 2008*). In addition, the required sensitivity is of the same order of magnitude as thermal fluctuations. Both cilia beating and the thermal fluctuations could theoretically be overcome by time averaging if mechanosensitive cilia acted as low-pass filters. The observed average velocities are still below the detection threshold found in other mechanosensitive cilia (*Delling et al., 2016*; *Goetz et al., 2014*; *Nauli et al., 2008*; *Rydholm et al., 2010*) and we are not aware of systems that detect static forces with such sensitivity. Hair bundles are known to achieve remarkable sensitivity thresholds close to the level of thermal noise, but the majority of them detect stimuli at higher frequencies and therefore act as high- or band-pass filters (*Muller et al., 2016*). A notable exception is the utricular otolith, which acts as a low-pass filter, but senses significantly higher forces (*Inoue et al., 2013*). We conclude that small flow velocities are a major challenge to the reliable sensing of directionality. Moreover, the hypothesis that cilia are capable of directional flow sensing remains untested in vivo (*Shinohara and Hamada, 2017*).

In addition, we showed that due to spatial inhomogeneities resulting from the cilia distribution, temporal averaging within a single cilium could not be sufficient for robust side determination via detection of flow direction. Signals from several cilia on each side would need to be ensemble-averaged. For robust flow sensing, we found that three cilia on each side are necessary.

We observed in vivo that the number of immotile cilia decreases over time along with an increase in the number of motile cilia, resulting in a stronger flow velocity and directionality. Nevertheless, the stage at which the number of immotile cilia is high corresponds to the stage where flow is the lowest and the least robust. At later stages, when it directionally is established, we observed that only 5% of cilia were immotile, less than three on each side. The number of immotile cilia is therefore too low to sense the flow direction. Combined with the related finding that cilia from isolated mouse cells do not bend significantly in response to flow applied at endogenous flow amplitudes (*Delling et al., 2016*), we conclude that a mechanism of symmetry breaking based solely on mechanical sensing of directional flow appears incompatible with our in vivo analysis.

These physical limitations led us to reconsider mechanosensing in the framework of LR symmetry breaking. Mechanosensing could, for instance, occur through the sensing of the cilia's own motion. The mean torque exerted by the fluid on a meridionally tilted beating cilium has the opposite direction from the torque exerted by the directional flow on an immotile cilium. A prediction of this auto-sensing hypothesis is therefore that an artificially induced counter-rotating flow (clockwise in dorsal view) would enhance, rather than reverse, the LR difference. Results in the mouse node (*Nonaka et al., 2002*) showing that externally imposed flow of opposite directionality can cause asymmetry reversal, contradict this view. Another observation in Xenopus that cannot be explained by the auto-sensing hypothesis is that the laterality mechanism breaks down when the fluid is made viscoelastic, which brings the directional flow to a halt, even though the cilia are still motile (*Schweickert et al., 2007*).

Conversely, our numerical simulations show that asymmetric transport of signaling molecules in the KV is a much more robust strategy than sensing mechanical cues. We characterize here the physical limit on the size of the signaling molecule for its reliable asymmetric distribution. Our calculations show that such mechanism requires the particle size to be bigger than 2 nm to work. Interestingly, this is in the size range of membrane-bound extra cellular vesicles (ECVs). ECVs play important roles in intercellular communication and may mediate a wide range of physiological and

pathological processes (*Cocucci et al., 2009*; *Hogan et al., 2009*; *Raposo and Stoorvogel, 2013*; *Wood et al., 2013*).

In summary, we analyzed the physical limits of mechanisms that have been proposed for asymmetry establishment in zebrafish. Combining large-scale in vivo imaging with fluid dynamics calculations we were able to map the biophysical features of cilia in the KV and the flows they generate in unprecedented detail. This allowed us to quantitatively test the physical limits of flow detection mechanisms. We show that the small number of immotile cilia found in the KV cannot be sufficient to robustly detect the direction of the flow given its high local variability. Motile cilia could sense the torques exerted by the fluid as a result of their own motion, which largely surpass the influence of the directional flow. However, this mechanism is incompatible with findings in other vertebrate species in which laterality establishment was suppressed in viscoelastic fluids and reversed with an artificial flow (*Nonaka et al., 2002*; *Schweickert et al., 2007*). Finally, we show that a chemosensory mechanism in which a LR gradient is established by combining directional flow around the dorsoventral axis with asymmetric particle secretion in the anterior region could explain the observed robust LR asymmetry establishment, provided that the particle size is above the lower limit of about 2 nm. Although the molecular nature of the flow detection mechanism remains obscure, our analysis of physical limitations of two proposed mechanisms rules out directional flow sensing. It also allows us to predict the minimum size of the signaling particle, which will eventually facilitate the search for it.

## Materials and methods

### Zebrafish strains

The zebrafish transgenic line used in the study is *actb2:Mmu.Arl13b*-GFP (*Borovina et al., 2010*). Embryos were raised at 32°C in the dark. For imaging, embryos were soaked in with Bodipy TR (Molecular Probes) for 60 min prior to the desired developmental stage and were subsequently embedded in 0.8% low melting point agarose (Sigma Aldrich) in Danieau solution. Embryos were imaged between 3- and 14-somite stages (SS).

### 2-photon excitation fluorescence (2PEF) microscopy

To image deep enough into the zebrafish embryo and capture the entire Kupffer's vesicle (KV), each live embryo (n = 28) was imaged using 2PEF microscopy with a TCP SP5 or SP8 direct microscope (Leica Inc.) at 930 nm wavelength (Chameleon Ultra laser, Coherent Inc.) using a low magnification high numerical aperture (NA) water immersion objective (Leica, 25x, 0.95 NA). We imaged the KV of embryos labeled with both *Arl13b*-GFP and BodipyTR between 3- and 14-SS: $100 \times 100 \times 50 \ \mu m^3$ 3D-stacks with $0.2 \times 0.2 \times 0.8 \ \mu m^3$ voxel size and 2.4 $\mu$s pixel dwell time were typically acquired to maximize the scanning artefact allowing to properly reconstruct cilia orientation in 3D (*Figure 1—figure supplement 1E*) as described in *Supatto and Vermot (2011)*. The fluorescence signal was collected using hybrid internal detectors at 493–575 nm and 594–730 nm in order to discriminate the GFP signal labeling cilia from the signal labeling the KV cell surface. To uncover the orientation of the KV within the body axes, the midline was also imaged. We typically imaged a volume of 600 $\times$ 600 $\times$ 150 $\mu m^3$ comprising the midline and the KV from top to bottom with a voxel size of 1.15 $\mu$m laterally and 5 $\mu$m axially.

### 3D-CiliaMap: quantitative 3D cilia feature mapping

We devised 3D-CiliaMap, a quantitative imaging strategy to visualize and quantify the 3D biophysical features of all endogenous cilia in the 50 to 80 cells constituting the KV in live zebrafish embryos from 3- to 14-SS (*Figure 1—figure supplement 1*). We used Imaris (Bitplane Inc., RRID:SCR_007370) and custom-made scripts in Matlab (The MathWorks Inc., RRID:SCR_001622) to perform image processing, registration, and analysis, and to extract the following features: KV size, shape and volume, cilia motility, number of cilia per KV, cilia spatial distribution, orientation of rotational axis and surface density (*Figure 1—figure supplement 1* and *Table 1*). Cilia motility, position, and orientation in 3D, as well as the reference frame of the body axes, were obtained from 2PEF images and exported from Imaris to Matlab using ImarisXT (*Figure 1—figure supplement 1D,E*). Since our analysis relies on the fluorescence signal from the cilia, we discarded embryos with levels of GFP expression too low to analyze them. Similarly, a few cilia per vesicle could be discarded when the signal or the

spatial resolution was too low to accurately determine motility (11%) or orientation (14%). Each cilium was defined as a unit vector from its base to its tip (*Figure 1—figure supplement 1E*). Cilia positions were registered in the body plan reference frame. To estimate the KV surface, we fitted an oblate spheroid to the distribution of cilia bases using the Ellipsoid fit Matlab script by Yury Petrov (Northeastern University, Boston, MA) (see fitting residues in *Table 1*). We used cilia vector components in the local orthogonal basis $(\vec{e}_f, \vec{e}_n, \vec{e}_m)$ defined at each cilium position on the spheroid surface to quantify cilia orientation angles θ (cilium tilt angle respective to the surface normal) and φ (cilium orientation on the KV cell surface), as shown in (*Figure 1C*). The experimental values were combined from different embryos and displayed in rosette histograms using Matlab. Finally, to estimate the local cilia density we transformed the spheroid into a sphere with surface density conservation and computed a spherical Voronoi diagram of cilia distribution based on the sphere_voronoi Matlab package by John Burkardt (Department of Scientific Computing, Florida State University, https://people.sc.fsu.edu/~jburkardt/). The scripts are available upon request.

## Flow calculation

When calculating the flow in a KV we first approximate it with a sphere of equal volume. We describe each cilium as a chain of 10 spheres (radius $a$=0.2µm) with a total length of $L$=6 µm, circling clockwise along a tilted cone with a frequency of 25 Hz. The phases were chosen randomly under the constraint that collisions between cilia were prevented (in rare cases, when inconsistencies in datasets led to unavoidable collisions, a randomly chosen cilium was removed). The mobility matrix M of the system was calculated using the Green's function for point forces inside a spherical cavity as described in *Maul and Kim (1994)*. For the diagonal elements (self-mobility of a particle) we used the expressions

$$\mathrm{M}_{ii}^{\mathrm{rr}} = \frac{1}{6\pi\eta a}\left(1 - \frac{9}{4}\frac{a}{R}\frac{1}{1-\vec{x}_i^2/R^2}\right) \qquad \mathrm{M}_{ii}^{\mathrm{tt}} = \frac{1}{6\pi\eta a}\left(1 - \frac{9}{8}\frac{a}{R}\left(\frac{1}{1-\vec{x}_i^2/R^2}+1-\frac{1}{2}\frac{\vec{x}_i^2}{R^2}\right)\right) \tag{3}$$

where $\mathrm{M}^{\mathrm{rr}}$ denotes radial and $\mathrm{M}^{\mathrm{tt}}$ the tangential mobility. $\vec{x}_i$ is the position of particle $i$ relative to the center of the KV. In each step the forces on the particles representing points on cilia were calculated by solving the linear equation system $\vec{v}_i = \sum_j \mathrm{M}_{ij}\vec{F}_j$ and the fluid velocities subsequently from the Green's function.

The 2D flow profiles (e.g., *Figure 4B*) were created by averaging the velocity over time, over cilia phases, as well as across a layer between $z = -0.1R$ and $0.1R$. The flow profiles along the AP axis (*Figure 5*) were averaged over time, phases and a region between $y = -0.05R \ldots 0.05R$ and $z = -0.05R \ldots 0.05R$.

As a simple and well-defined measure to characterize the intensity and directionality of the flow, we introduced the effective angular velocity

$$\vec{\Omega} = \frac{5}{2VR^2}\int \vec{x}\times\vec{v}(\vec{x})\,dV, \tag{4}$$

that is, the angular velocity of a uniformly rotating sphere with the same angular momentum as the fluid in the KV (shown in *Figure 4C*). Note that the angular momentum is used solely as a velocity measure since the fluid inertia is negligible. $\vec{\Omega}$ can be calculated directly from the force distribution as

$$\vec{\Omega} = \frac{3}{16\pi\eta R^3}\sum_i\left(1-\vec{x}_i^2/R^2\right)\vec{x}_i\times\vec{F}_i, \tag{5}$$

thus omitting the need for spatial integration. This non-trivial expression can be derived from the following considerations. From symmetry arguments, we know that the effective angular velocity caused by a point force $\vec{F}$ acting at point $\vec{x}$ inside the cavity can only have the form

$$\vec{\Omega} = w(|\vec{x}|)\,\vec{x}\times\vec{F} \tag{6}$$

with an unknown scalar function $w(r)$. We now consider a distribution of forces on a concentric sphere with radius $r_i$ such that the velocity inside is

$$\vec{v}(\vec{x}) = \vec{\Omega}_0 \times \vec{x} \begin{cases} 1, & |\vec{x}| \leq r_i \\ \frac{|\vec{x}|^{-3} - R^{-3}}{r_i^{-3} - R^{-3}}, & |\vec{x}| > r_i \end{cases}. \tag{7}$$

Using definition (4), the effective angular velocity of this distribution can be obtained by spatial integration with the result

$$\vec{\Omega} = \vec{\Omega}_0 \frac{3}{2} \left(1 - \frac{r_i^2}{R^2}\right)\left(\frac{R^3}{r_i^3} - 1\right)^{-1}. \tag{8}$$

At the same time, the force density at the inner sphere that maintains the velocity profile (7), is

$$\vec{f} = \vec{\Omega}_0 \times \vec{x} \, \frac{3\eta}{r_i} \left(1 - \frac{r_i^3}{R^3}\right)^{-1}. \tag{9}$$

Inserting this force density into *Equation (6)* and integrating over the inner sphere gives

$$\vec{\Omega} = \vec{\Omega}_0 w(r_i) 8\pi\eta R^3 \left(\frac{R^3}{r_i^3} - 1\right)^{-1}. \tag{10}$$

The expressions (8) and (10) become equivalent when $w(r_i) = \frac{3}{16\pi\eta R^3}\left(1 - \frac{r_i^2}{R^2}\right)$, which leads to the *Equation (5)* for the effective angular velocity.

## Randomly generated vesicles

In addition to the available datasets, we extended our analysis to randomly generated cilia distributions that shared the main features with those observed in real KV. The vesicles were assigned a radius $R = 35\,\mu$m. We randomly distributed $N_c = 70$ cilia with a density function $\rho$ that had its maximum at $(\alpha, \beta) = (0, 15°)$ and a pair correlation function $g(\gamma)$ resembling the measured one (*Figure 4C*). All cilia were tilted meridionally (towards dorsal) with a tilt angle $\theta = 60° \times \cos\beta$. The randomly generated distributions allowed us to study the reliability of the proposed flow sensing hypotheses without being limited by the number of vesicles analyzed experimentally. In each simulation with randomly generated vesicles, 200 vesicles were simulated in order to obtain stable results.

## Torques

To test the mechanosensing mechanism, we chose three immotile cilia situated in the left region of the KV (up to 45° away from the left pole), and three cilia situated in the right region of the KV (up to 45° away from the right pole). In case the number of immotile cilia in one region was insufficient we randomly assigned additional cilia as immotile. To test the case with maximum sensitivity, the passive cilia were set normal to the surface ($\theta = 0$). After determining the forces on all cilia, the meridional component (in the direction of $\vec{e}_m$ on the left and $-\vec{e}_m$ on the right) of the torque vector was evaluated around the base of a cilium. *Figure 7B,C* shows the cumulative distributions of instantaneous values of these torques, their temporal average, as well as the temporal and ensemble average for a group of 3 cilia on each side.

## Particle diffusion

We evaluated the model based on diffusion of signaling particles with a Langevin-dynamics simulation in the fluid velocity field evaluated before (*Video 3*). We assumed that the particles are secreted from random points in a region 30° around anterior and captured whenever they encounter a cilium elsewhere. The simulation step was 0.001 s and the number of particles traced 1000. The diffusion constant of a particle was determined as $D = k_B T/(6\pi\eta\, r_{\text{Stokes}})$ with the fluid viscosity $\eta = 0.001\,\text{Pa}\,\text{s}$. Particles captured by cilia in the left and right region (up to 45° away from the left/right direction) were counted and the average ratio $N_{\text{left}}/N_{\text{left+right}}$, as well as its 5th and 95th percentile were plotted.

## Statistical analyses

We did not compute or predict the number of samples necessary for statistical differences because the standard deviation of our study's population was not known before starting our analysis. Biological replicate corresponds to the analysis of different embryos of the same stage. Technical replicate corresponds to the analysis of the same embryo imaged the same way. The sample size (replicate and number) to use was as defined by our ability to generate our datasets. We routinely analyze 5 to 10 embryos at each considered stage. Both the mean and the SD (*Figure 4—figure supplement 1* and *Table 1*) were calculated for several of the KV and cilia features measured. For analyses between two groups of embryos, differences were considered statistically significant when the p-value<0.05, as determined using a two-tailed and paired Student's t-test. Circular statistics (resultant vector length $r$ and 95% confidence intervals on the estimation of the mean angle) where computed using the CircStat Matlab toolbox (*Berens, 2009*). Descriptive statistics (cilia density maps) were displayed using Matlab custom scripts (the scripts are available upon request).

## Acknowledgements

We thank C Wyart, C Norden, M Blum, D Riveline, E Beaurepaire, G Pakula and the Vermot lab for discussion and thoughtful comments on the manuscript, in particular R Chow for her help with editing. We thank the Lopes group and the Furthauer group for sharing data prior to publication. We thank B Ciruna for providing fish stocks. We thank the IGBMC fish facility (S. Geschier and S. Gredler) and the IGBMC imaging center, in particular B Gurchenkov, P Kessler, M Koch and D Hentsch. This work was supported by HFSP, INSERM, AFM, FRM (DEQ20140329553), the European seventh framework program (MC-IRG256549 and MC-IRG268379), ANR (ANR-12-ISV2-0001, ANR-11-EQPX-0029, ANR-2010-JCJC-1510–01) and by the grant ANR-10-LABX-0030-INRT, a French State fund managed by the Agence Nationale de la Recherche under the frame program Investissements d'Avenir labeled ANR-10-IDEX-0002–02. R.R.F. was supported by the IGBMC International PhD program (LABEX). AV acknowledges support from the Slovenian Research Agency (grant J1-5437).

## Additional information

### Competing interests

FJ: Reviewing editor, *eLife*. The other authors declare that no competing interests exist.

### Funding

| Funder | Grant reference number | Author |
| --- | --- | --- |
| Human Frontier Science Program | CDA00032/2010-C | Julien Vermot |
| Labex | ANR-10-LABX-0030-INRT | Rita R Ferreira Julien Vermot |
| Agence Nationale de la Recherche | ANR-13-BSV1-0016 | Julien Vermot |
| Agence Nationale de la Recherche | ANR- 12-ISV2-0001 | Julien Vermot |
| Agence Nationale de la Recherche | ANR-2010-JCJC-1510-01 | Willy Supatto |
| Agence Nationale de la Recherche | ANR-11-EQPX-0029 | Willy Supatto |
| Javna Agencija za Raziskovalno Dejavnost RS | J1-5437 | Andrej Vilfan |

The funders had no role in study design, data collection and interpretation, or the decision to submit the work for publication.

## Author contributions

RRF, Data curation, Formal analysis, Investigation, Visualization, Writing - review and editing; AV, Conceptualization, Data curation, Software, Formal analysis, Funding acquisition, Validation, Investigation, Visualization, Methodology, Writing - original draft, Writing - review and editing; FJ, Conceptualization, Formal analysis, Methodology, Writing - original draft, Writing - review and editing; WS, Conceptualization, Software, Formal analysis, Supervision, Funding acquisition, Investigation, Methodology, Writing - original draft, Writing - review and editing; JV, Conceptualization, Resources, Data curation, Supervision, Funding acquisition, Validation, Investigation, Writing - original draft, Project administration, Writing - review and editing

## Author ORCIDs

Rita R Ferreira, http://orcid.org/0000-0001-7291-9495
Andrej Vilfan, http://orcid.org/0000-0001-8985-6072
Frank Jülicher, http://orcid.org/0000-0003-4731-9185
Willy Supatto, http://orcid.org/0000-0002-4562-9166
Julien Vermot, http://orcid.org/0000-0002-8924-732X

## Ethics

Animal experimentation: Animal experiments were approved by the Animal Experimentation Committee of the Institutional Review Board of the IGBMC.

# Additional files

## Supplementary files

• Source code 1. Matlab script describing the structure of KVdata.mat information (see script comments) and displaying a sample figure of cilia distribution in a vesicle to show how to use this MAT-file.

• Source code 2. "KVdata.mat" is a MAT-file containing all vesicle features computed in this study (see Source code 1 for information about its content and use).

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

## Appendix A

### Derivation of the fluid velocity above a cilia layer with a density gradient

In order to derive the expression for the fluid velocity above an inhomogeneous layer of short cilia (**Equation 2**), we look into the equivalent problem with a planar boundary condition. A symmetrically rotating cilium (no tilt) located at $\vec{X}$ is surrounded by a vortical flow with a far-field velocity profile

$$\vec{v}\left(\vec{x}\right) = \frac{3C_N\omega L^4}{16\pi\eta}\sin^2(\psi)\cos(\psi)z\,\frac{\vec{x}-\vec{X}}{|\vec{x}-\vec{X}|^5}\times\vec{e}_n \tag{11}$$

where $z$ is the height above the surface (**Vilfan, 2012**). A cilia carpet with surface density $\rho(\vec{X})$ then produces the flow

$$\vec{v}\left(\vec{x}\right) = \frac{C_N\omega L^4}{16\pi\eta}\sin^2(\psi)\cos(\psi)\int\rho\left(\vec{X}\right)z\left(\vec{\nabla}_X\frac{1}{|\vec{x}-\vec{X}|^3}\right)\times\vec{e}_n\,dS \tag{12}$$

By applying partial integration the expression can also be written as

$$\vec{v}\left(\vec{x}\right) = -\frac{C_N\omega L^4}{16\pi\eta}\sin^2(\psi)\cos(\psi)\int\vec{\nabla}\rho\left(\vec{X}\right)\times\vec{e}_n\frac{z}{|\vec{x}-\vec{X}|^3}\,dS \tag{13}$$

For a small $z$, the integral yields $2\pi\vec{\nabla}\rho\times\vec{e}_n$ and we obtain **Equation (2)**. Because we assumed $L \ll R$, the derivation is equally valid for a non-planar (e.g., spherical) boundary condition. Thus, the effective slip velocity only depends on the density gradient. An infinite surface, uniformly lined with rotating cilia, does not produce any far-field flow.

## Appendix B

### Thermal noise on an elastic cilium

To estimate the thermal noise on a cilium, we treat it as an elastic beam with a flexural rigidity of $EI = 3 \times 10^{-23} \mathrm{Nm^2}$ (**Battle et al., 2015**). For cilia of this length, the dynamics is dominated by the fundamental bending mode and the cilium can be treated as a damped harmonic oscillator. The tip of the cilium then acts as an elastic spring with a spring constant $K = 3\,EI/L^3$ . From the equipartition theorem it follows that the r.m.s. tip deflection is $\sqrt{k_B T/K}$, which corresponds to a torque measured at the base of the cilium ($\tau = K\,L\,\sqrt{k_B T/K} = \sqrt{3 k_B T\,EI\,/\,L} = 2.5 \times 10^{-19}$ Nm). The relaxation rate of the cilium is $\Gamma = EI\,k_1^4/(C_N\,L^4)$ with $k_1 \approx 1.89$ (**Battle et al., 2015**), which gives $\Gamma = 80\ \mathrm{s^{-1}}$ and also determines the corner frequency of thermal noise. The spectral density of the force fluctuations is therefore

$$\langle \tau^2(\omega) \rangle = \frac{2}{\pi}\,k_B T\,K L^2\,\frac{\frac{1}{\Gamma}}{1 + \frac{\omega^2}{\Gamma^2}}. \tag{14}$$

A low pass filter with a time constant $T$ reduces the spectral density at frequency $\omega$ by a factor $1/\left(1 + (\omega T)^2\right)$. The total noise amplitude after filtering is given by the integral over the frequency spectrum

$$\langle \tau_f^2 \rangle = \int_0^\infty \frac{\langle \tau^2(\omega) \rangle}{1 + (\omega T)^2}\,d\omega = \frac{\langle \tau^2 \rangle}{1 + \Gamma T}. \tag{15}$$

To reduce the detected r.m.s. amplitude of thermal noise to $2 \times 10^{-20}$ Nm, well below the proposed threshold for flow sensing, a time constant $T = 2$ s is necessary.

In a similar way, we can estimate the effect of low-pass filtering on the noise that is caused by the beating of adjacent cilia. The flow calculation on mid- and late stage KV yields an average r.m.s. amplitude of the oscillatory torque $\tau_{\mathrm{osc}} = 9 \times 10^{-19}$ Nm, mostly with the ciliary beating frequency $\omega_0 = 2\,\pi \times 25\ \mathrm{s^{-1}}$ (although higher harmonics are present). After filtering, the amplitude is reduced to $\tau_{\mathrm{osc-f}} = \tau_{\mathrm{osc}}/\sqrt{1 + (\omega_0 T)^2}$. The filtered amplitude can be brought down to the same level ($\tau_{\mathrm{osc-f}} = 2 \times 10^{-20} \mathrm{Nm}$) with a time constant $T = 0.3$ s.

The estimate shows that temporal averaging with a time constant longer than 2 s suppresses both the thermal and the oscillatory noise well below the estimated detection threshold. Spatial variability of the flow, on the other hand, still requires averaging over several immotile cilia.

