## [Decision Letter]

Thank you for submitting your article "Physical limits of flow sensing in the left-right organizer" for consideration by *eLife*. Your article has been reviewed by two peer reviewers, and the evaluation has been overseen by a Reviewing Editor and Didier Stainier as the Senior Editor. The following individual involved in review of your submission has agreed to reveal his identity: David E Clapham (Reviewer #2).

The reviewers have discussed the reviews with one another and the Reviewing Editor has drafted this decision to help you prepare a revised submission.

Summary:

This is an excellent and thorough characterization of fluid flow and mechanical forces in the zebrafish embryonic node. Provided that the sample is representative of KVs, and that time/spatial resolution was sufficient to accurate to quantify motion, the work shows that the chemosensory gradient mechanism in the node is the most likely LR organization mechanism. The work sets an excellent standard for quantification and rigorous mechanistic analysis for this important problem.

Essential revisions:

Regarding the presentation of the theoretical results:

i) Present the experimentally determined ciliary densities earlier in order to justify (if possible!) the slip-velocity approach. Since the authors rule in/out different scenarios based on the measured ciliary densities and the slip velocity model it is important to explain the limitations of that model.

The authors state above Eq. 1 that the justification for a slip velocity is "dense short cilia (i.e. ciliary length (*L*) and characteristic interciliary distance that are much shorter than the radius of the KV (*R*)." If we call the interciliary distance d I would have thought that not only should *L*<<*R* and *d*<<*R* but also d significantly less than L (the meaning of dense). Is this actually true in the KV? According to Figure 4 the typical surface densities are on the order of 0.01/μm^2^, so their typical spacing is 10 μm. The typical length assumed is 6 μm. This is not exactly a dense carpet of cilia.

ii) Improve the discussion about the mechanosensing mechanism.

The specific model for torque computation needs to be stated in the text, and some basic idea of the scale of flow is needed. How do these flow speeds compare with the tip velocity of the cilium? There is also the (unstated) issue of the timescale for averaging (a la Berg and Purcell). Can the problems of thermal noise be eliminated if the averaging were done over longer times?

iii) Improve the discussion about flow mediated transport of a secreted signaling molecule:

Is the situation envisioned one of steady release and absorption, or does the concentration build up significantly over time? The results presented are rather anecdotal and not particularly thorough.

iv) Reference the seminal work of Cartwright et al. [PNAS 101, 7234 (2004)] that was the first to establish theoretically various features of the large-scale flow from beating cilia.

---

## [Author Response]

*Essential revisions:*

*Regarding the presentation of the theoretical results:*

*i) Present the experimentally determined ciliary densities earlier in order to justify (if possible!) the slip-velocity approach. Since the authors rule in/out different scenarios based on the measured ciliary densities and the slip velocity model it is important to explain the limitations of that model.*

*The authors state above Eq. 1 that the justification for a slip velocity is "dense short cilia (i.e. ciliary length (L) and characteristic interciliary distance that are much shorter than the radius of the KV (R)." If we call the interciliary distance d I would have thought that not only should L<<R and d<<R but also d significantly less than L (the meaning of dense). Is this actually true in the KV? According to Figure 4 the typical surface densities are on the order of 0.01/μm^2^, so their typical spacing is 10 μm. The typical length assumed is 6 μm. This is not exactly a dense carpet of cilia.*

We agree with the reviewers about the limitations of the analytical (surface slip) model, whose main purpose is to gain a qualitative understanding of the mechanism behind rotational flow generation (meridional tilt vs. dorsal gradient). The conditions that need to be met for its validity both depend on the distance of the observed flow from the wall: it has to be large relative to the length of the cilia, as well as to the characteristic distance between cilia. In this limit the condition *d*<<*L* (dense carpet) is not necessary. In our case, it means that the approximation works reasonably well in the center of the KV, but not close to the walls. The numerical calculations shown in the last part of the manuscript (Figure 6 and its supplement) confirm this. We have rewritten the second paragraph of "Results" to clarify this.

*ii) Improve the discussion about the mechanosensing mechanism.*

*The specific model for torque computation needs to be stated in the text, and some basic idea of the scale of flow is needed. How do these flow speeds compare with the tip velocity of the cilium?*

We have rewritten and expanded the Results section (Mechanosensory mechanism 2). As we now state in the text, the tip of a cilium moves with 400 µm/s, much faster than the flow of ~10 µm/s. More about the torque calculation is written in Material and Methods, section "Torques".

*There is also the (unstated) issue of the timescale for averaging (a la Berg and Purcell). Can the problems of thermal noise be eliminated if the averaging were done over longer times?*

We have added a new Appendix B addressing the properties of noise and the ability to eliminate it by averaging (low-pass filtering). The estimate shows that both the thermal and the noise caused by adjacent beating cilia can be suppressed when averaged with a time constant longer than 2 seconds, which is not a limitation as the time scales of asymmetry establishment are much slower. We have also rewritten the relevant passage (Results, "Mechanosensory mechanism 1") to stress that spatial variations, along with the paucity of immotile cilia, provide the main argument against mechanodetection in the KV.

*iii) Improve the discussion about flow mediated transport of a secreted signaling molecule:*

*Is the situation envisioned one of steady release and absorption, or does the concentration build up significantly over time?*

The concentration profiles shown in Figure 7 are calculated for steady state release and absorption. The typical lifetime of secreted particles before absorption is of the order of 10 s. This implies that the steady-state distribution is reached quickly and there is no concentration buildup. However, we expect that the time course of secretion does not play a role in the actual asymmetry establishment, which we assume only depends on the cumulative numbers of particles detected on each side. We have clarified these points in the Results section (Flow-mediated transport of a signaling molecule).

*The results presented are rather anecdotal and not particularly thorough.*

We agree that there remain many unknowns about the proposed chemosensing mechanism and we had to use some assumptions to come to our conclusions. Our rationale is that the feasibility of a mechanism can be shown using a single plausible example and parameter set. Besides that, our results were quite robust when varying the assumptions regarding secretion and absorption (because Figure 7 is already quite complex, we have decided to not include these controls in the paper).

*iv) Reference the seminal work of Cartwright et al. [PNAS 101, 7234 (2004)] that was the first to establish theoretically various features of the large-scale flow from beating cilia.*

Cartwright's paper is now referenced in the Introduction and Results when the connection between the cilia tilt and the flow is mentioned.